# EL-Clustering: Combining Upper- and Lower-Bounded Clusterings for Equitable Load Constraints

**Rajni Dabas**[*]                                                                                      *rajni.dabas@pgdav.du.ac.in*
*Department of Computer Science, Northwestern University*
*P.G.D.A.V. College, University of Delhi*

**Neelima Gupta**[†]                                                                                    *ngupta@cs.du.ac.in*
*Department of Computer Science, University of Delhi*

**Rudra Bhardwaj**                                                                                   *rudramsc24@cs.du.ac.in*
*Department of Computer Science, University of Delhi*

**Sapna Grover**                                                                                          *sgrover@cs.du.ac.in*
*Department of Computer Science, University of Delhi*
*Dyal Singh College, University of Delhi*

**Reviewed on OpenReview:** *https://openreview.net/forum?id=EkjDfnJ1gU*

## Abstract

The application of an ordinary clustering algorithm may yield a clustering output where the number of points per cluster (cluster size) varies significantly. In settings where the centers correspond to facilities that provide a service, this can be highly undesirable as the cluster size is essentially the service load for a facility. While prior work has considered imposing either a lower bound on the cluster sizes or an upper bound, imposing both bounds simultaneously has seen limited work, especially for the $k$-median objective, despite its strong practical motivation. In this paper, we solve the *equitable load* (**EL**) clustering problem where we minimize the $k$-median objective subject to the cluster sizes not exceeding an upper bound or falling below a lower bound. We solve this problem using a modular approach. Specifically, given a clustering solution that satisfies the lower bound constraints and another that satisfies the upper bound constraints, we introduce a combination algorithm which essentially combines both solutions to produce one that satisfies both constraints simultaneously at the expense of a bounded degradation in the $k$-median objective and a slight violation of the upper bound. Our combination algorithm runs in $O(k^3 + n)$ time, where $n$ is the number of points and is faster than standard $k$-median algorithms that satisfy either the lower or upper bound constraints. Interestingly, our results can be generalized to various other clustering objectives, including the $k$-means objective. We also do empirical evaluation for $k$-Median objective on benchmark datasets to show that both, the cost as well as the violation factor are significantly smaller in practice than the theoretical worst-case guarantees.

## 1 Introduction

Decision-making using algorithms powered by machine learning has become ubiquitous. Routinely, algorithms are used in consequential applications such as loan approval (Sheikh et al. (2020); Kadam et al. (2021)), recidivism prediction (Travaini et al. (2022); Kovalchuk et al. (2023)), and kidney exchange (Ashlagi & Roth (2021); McElfresh et al. (2020)). This has naturally brought greater attention to the broader

---

[*]This work was done when the auhtor was a research scholar at Department of Computer Science, University of Delhi.
[†]Corresponding author.

impact of these deployed algorithms and their vulnerability to noise and adversarial attacks, as well as their societal consequences in terms of fairness and privacy. These additional considerations imply that there is a real need to solve non-standard variants of many problems to overcome these possible harmful consequences.

Variants of the standard clustering problem that take such considerations into account have received significant attention from the research community; this is unsurprising since clustering is a fundamental problem in unsupervised learning and a classical problem in operations research. Examples of such works include Jones et al. (2021); Gupta et al. (2010); Kaplan & Stemmer (2018), who show algorithms for solving the $k$-median and $k$-means problems that preserve the privacy of individuals using differential privacy. Further, Chhabra et al. (2020) and Cinà et al. (2022) study the performance of clustering algorithms when the dataset is affected by adversarial corruptions. Moreover, fairness considerations in clustering have received even greater attention comparatively (Chierichetti et al. (2017); Bercea et al. (2019); Bera et al. (2019); Kleindessner et al. (2019); Ahmadi et al. (2022); Chen et al. (2019); Li et al. (2021); Chakrabarti et al. (2022); Awasthi et al. (2022)[1]). The fair clustering literature has introduced a number of well-motivated fairness notions. At least seven different fairness notions have been introduced so far in clustering.

Despite the significant attention that has resulted in many variants of the classical clustering problem, we identify a simple notion that has not received much attention from the community, even though it is well-motivated and has clear societal consequences. Specifically, from the operations research point of view, the selected centers in a clustering could represent facilities such as schools, with the cluster associated with each center (school) being the students assigned to that school. Naturally, a school requires a minimum number of students to maintain a good teaching quality[2] at the same time the number of students should not exceed a certain threshold as the school's resources might be over-consumed leading to a degradation in the teaching quality. One can also find a similar motivation if the schools were instead service centers providing services to clients instead of students. Each service center would want a minimum number of clients to bring in revenue; at the same time, the number of clients should not exceed a threshold, as that would lead to issues such as higher waiting time and lower service quality. At a more precise level, this notion, which we call *equitable service load (EL)*, simply states that the size of each cluster (number of points in the cluster) should be both lower and upper bounded by some pre-set values simultaneously. This **EL** notion can also be motivated in machine learning applications such as market segmentation. Specifically, since points in the same cluster would receive the same ads, we might want to have a level of equity between the different ads (centers) so that none receive too little revenue or dominate the market.

Though clustering under **EL** constraints has not received much attention, this notion is not entirely new. Specifically, the literature has considered variants of the standard clustering problem where lower and upper bounds on the cluster sizes have to be satisfied simultaneously (Friggstad et al. (2016); Gupta et al. (2021); Ding et al. (2017); Rösner & Schmidt (2018)). However, the $k$-median variant of this problem (with lower and upper bounds) remains unsolved. Only heuristics have been introduced for the stringent case where the lower and upper bounds coincide (i.e., set to the same value); see de Maeyer et al. (2023) and citations within. Forcing the upper and lower bounds to be exactly equal is not practical since in most settings a small difference would be tolerated even if an exact equality was desired.

In this paper, we solve the $k$-median problem under **EL** constraints. Unlike the prior work, we follow a modular approach. Specifically, using a solution where the cluster sizes are all lower bounded and another where the cluster sizes are upper bounded, we introduce a post-processing algorithm that combines the two to give a new solution that satisfies the **EL** constraints. Our combination algorithm runs in $O(k^3 + n)$ time, where $n$ is the number of points, and as such does not present a heavy computational burden. In fact, in comparison to existing algorithms for the $k$-median problem subject to either a lower or upper bound constraint on the cluster size, our combination algorithm does not present the computational bottleneck in the algorithmic pipeline. Interestingly, although our main target is the $k$-median problem, we show how we can use our combination algorithm to solve other clustering variants under the **EL** constraints, including $k$-means clustering.

---

[1]See the references therein for more.

[2]It is well-known and documented that interaction between students can improve the educational and social outcomes (Soller (2001); Hurst et al. (2013)) but this would not be possible with a very small student body.

**Organization of the Paper.** In Section 2, we give our notation along with the formal statement of the **EL** problem and some background on relevant prior work. In Section 3, we state our main theoretical results. In Section 4, we give an overview of additional related work to our problem. In Section 5, we give a high-level discussion of our main algorithmic techniques. In Section 6, we present our algorithm for **EL** Clustering for the $k$-median objective along with detailed technical proofs that establish the guarantees of the algorithm. This is followed by a conclusion and a discussion of future work in Section 7. In addition, Appendix A discusses improvements to some of the theoretical guarantees when the gap between the lower and upper bounds is sufficiently large. Appendix B presents the algorithmic modifications required for other variants of the problem, while Appendix C contains the experimental setup and empirical evaluation.

## 2   Notation, Problem Statement, and Background

In our problem, we are given a set of locations $P$ in a metric space with metric $c : P \times P \to \mathbb{R}_{\geq 0}$, a subset $\mathcal{C} \subseteq P$ of $n$ many points to be clustered. Further, we are given a set of (potential) centers[3] $\mathcal{F} \subseteq P$ and a positive integer $k$. Following the standard terminology in clustering and facility location, we will also refer to the given $n$ points $\mathcal{C}$ as *clients* and to the centers $\mathcal{F}$ as *facilities*. As in standard $k$-median clustering, our objective is to find a set of facilities $\mathcal{F}' \subseteq \mathcal{F}$ of at most $k$ facilities (i.e., $|\mathcal{F}'| \leq k$, note that this is called the *cardinality constraint*) and an assignment function $\sigma : \mathcal{C} \to \mathcal{F}'$ which assigns clients to the selected facilities so as to minimize the sum of distances between the clients and their assigned facilities. More formally, we want to obtain a solution $S = (\mathcal{F}', \sigma)$ that minimizes the objective function $Cost(S) = \sum_{j \in \mathcal{C}} c(j, \ \sigma(j))$. Furthermore, in **EL** Clustering we are additionally given two parameters $L$ and $U$ where $L$ is a lower bound on the cluster size and $U$ is the upper bound. It follows that in a valid **EL** Clustering, the size of any cluster is constrained to lie in the range $[L, U]$. More precisely, denoting the set of points assigned to a center $i \in \mathcal{F}'$ by $\sigma^{-1}(i)$, it follows that $L \leq |\sigma^{-1}(i)| \leq U$. From the above description, the formal and concise definition of the **EL** Clustering problem is as follows:

---

> **EL** Clustering
>
> **Input:** Instance $I = (P, c, \mathcal{C}, \mathcal{F}, k, U, L)$
> **Optimization:** $\min_{\mathcal{F}', \sigma} \sum_{j \in \mathcal{C}} c(j, \ \sigma(j))$   subject to
>
> - subset $\mathcal{F}' \subseteq \mathcal{F}$ of size at most $k$,
>
> - assignment $\sigma : \mathcal{C} \to \mathcal{F}'$ such that for each facility $i \in \mathcal{F}'$, $L \leq |\sigma^{-1}(i)| \leq U$

---

We use three standard notions from the literature on approximation algorithms:

1. **Approximation Factor.** An algorithm is said to have an $\alpha$-*approximation factor* if, for every instance, the cost of the solution it produces is at most $\alpha$ times the cost of an optimal solution.

2. **Violation Factor.** A solution is said to *violate* the upper bound constraint by a *factor* $\beta \geq 1$ if the solution exceeds the constraint by at most a multiplicative factor of $\beta$. For example, in **EL** clustering, a violation factor $\beta$ means that no cluster has more than $\beta U$ clients.

3. **Bi-criteria Approximation.** A solution is an $(\alpha, \beta)$-*bi-criteria approximation* if it returns a solution of cost at most $\alpha$ times optimal and violates upper bound constraint by at most a factor of $\beta$.

We next define two $k$-MEDIAN problems satisfying partial **EL** constraints (dropping one of the two, lower or upper bound constraints) whose solutions will be combined to obtain a solution for **EL** Clustering. In UPPER BOUNDED $k$-MEDIAN (**U**$k$**M**), we drop the lower bounds (which is equivalent to setting $L = 0$), whereas in LOWER BOUNDED $k$-MEDIAN (**L**$k$**M**), we drop the upper bounds (by setting $U = n$). Both of these problems have been extensively studied in the theoretical computer science literature. Though interesting, upper bound constraints are notoriously hard to handle in problems like $k$-median. For example, finding a constant factor approximation for **U**$k$**M** is one of the famous and long-standing open questions in the literature of

---

[3]Note that it is common in the classical $k$-median clustering to select the centers from the same set of points to be clustered, this can be captured in our formulation by simply setting setting $\mathcal{F} = \mathcal{C}$.

approximation algorithms. On the other hand, some heuristics do not provide any approximation ratio guarantees (i.e., a bound on the cost of the solution as compared to the cost of the optimal solution). On the positive side, there are several papers that solve the problem by giving bi-criteria approximations that have an approximation ratio for the clustering objective, but also violate either the upper bounds or the cardinality constraint by a small multiplicative factor (Byrka et al. (2015; 2016); Charikar et al. (1999); Demirci & Li (2016); Korupolu et al. (2000); Li (2014; 2015; 2016)). For the **L$k$M** problem, both constant factor approximations and heuristics have been obtained (Arutyunova & Schmidt (2021); Guo et al. (2020); Han et al. (2020b;a)).

## 3  Our Results

In this paper, we study **EL** Clustering to obtain Theorem 3.1.

**Theorem 3.1.** *Given a solution $S_U$ for UPPER BOUNDED $k$-MEDIAN (U$k$M) with an upper bound violation of factor $\beta$ and a solution $S_L$ for LOWER BOUNDED $k$-MEDIAN (L$k$M). If the clustering costs of the solutions are $Cost(S_U)$ and $Cost(S_L)$, respectively. Then a solution of cost at most $(7Cost(S_U) + 2Cost(S_L))$ can be obtained for **EL** Clustering at a violation of the upper bound by a factor of $(\beta + 1)$ and in a run-time of $O(k^3 + n)$.*

Note that the $O(k^3 + n)$ runtime stated above refers only to the *combination step* that merges the given solutions $S_U$ and $S_L$; it does not include the time required to compute $S_U$ and $S_L$, which depends on the algorithms used for UPPER BOUNDED $k$-MEDIAN and LOWER BOUNDED $k$-MEDIAN. The theorem also establishes bounds on the clustering cost and the violation in the upper bound constraints using any two given solutions, even if these solutions result from heuristics. It follows that if we obtain these solutions using approximation algorithms for UPPER BOUNDED $k$-MEDIAN and LOWER BOUNDED $k$-MEDIAN, then we can establish approximation ratio guarantees for our **EL** solution as shown in the corollary below:

**Corollary 3.2.** *Given an $\alpha_U$ approximation for UPPER BOUNDED $k$-MEDIAN with $\beta$ violation in the upper bounds and an $\alpha_L$ approximation for LOWER BOUNDED $k$-MEDIAN, a $7\alpha_U + 2\alpha_L$ approximation can be obtained for **EL** Clustering with a $(\beta + 1)$ violation in upper bounds in $O(k^3 + n)$ time.*

Applying existing algorithms in Corollary 3.2 immediately yields concrete bounds: using the 16-approximation with 3-factor upper bound violation of Charikar et al. (1999) for **U$k$M** and the 387-approximation for **L$k$M** of Han et al. (2020a), we obtain an 886-approximation with a 4-factor violation. Alternatively, with the $O(1/\varepsilon^2)$-approximation of Byrka et al. (2016) for **U$k$M** and small $\varepsilon > 0$, the violation factor approaches 2. The constants here are driven by the underlying algorithms for **U$k$M** and **L$k$M**, and any future improvements in these directly improve our guarantees.

Moreover, our combination approach works independently of the specific technique used in these algorithms; the result holds even if the input solutions are produced by heuristics without formal guarantees, in which case our algorithm preserves their practical performance. For instance, by applying the Fixed Parameter Tractable (FPT) approximation algorithms[4] for **U$k$M** and **L$k$M** by Goyal et al. (2020), the approximation factor for **EL** clustering can be reduced to $(27+\epsilon)$ with a 2-factor violation in upper bounds, while maintaining an FPT runtime in $k$.

Furthermore, the computational overhead incurred by our algorithm in combining the solutions is only $O(k^3 + n)$ whereas all algorithms[5] for **U$k$M** and **L$k$M** require solving a linear programming problem and hence takes at least $\omega(n^4)$ time (Vaidya (1989); Jiang et al. (2020); Cohen et al. (2021); van den Brand (2020)).

Interestingly, our technique can be extended to other clustering variants, such as $k$-MEANS, $k$-CENTER, FACILITY LOCATION and KNAPSACK MEDIAN in the presence of **EL** constraints. Although we are able to extend the result to the $k$-Means problem, the constants associated with the cost of generating a $k$-means clustering with equitable load are relatively high in our paper. Improving these constants remains an

---

[4] An algorithm is FPT if its runtime is upper bounded by $O(f(k) \cdot n^c)$ where $c$ is a constant but $f(k)$ can be exponential in $k$, see Cygan et al. (2015) for more details.

[5] Except Korupolu et al. (2000) that uses local search but opens $(5 + \epsilon)k$ facilities instead of $k$.

interesting open question for future work. We mainly focus on $k$-median in the paper; modifications in the algorithm for other problems can be found in the supplementary material (Appendix B).

In the supplementary material (Appendix A), we show an improvement in the upper bound violation for a particular scenario when the gap between the lower and the upper bounds is not too small, specifically when $2L \leq U$. Note that this is a reasonable scenario that is likely to occur in real applications. For this special case, we reduce the violation in the upper bounds to $(\beta + \epsilon)$ at the expense of an increase of a factor of $O(1/\epsilon)$ in the cost for a constant $\epsilon > 0$.

Finally, we complement our theoretical guarantees with an empirical evaluation (Section C) on benchmark datasets, showing that the cost as well as the violation factor are substantially better than the worst case guarantees in most of the cases. It is observed that the cost overhead is within 20% of the cost of maximum of the **L**$k$**M** and **U**$k$**M** solutions and it is typically within 10–11%, whereas the violation factor is within 1.54 in $\approx 80\%$ of the cases.

## 4 Additional Related Work

As mentioned earlier, **EL** Clustering has not received much attention from the community. Heuristics are known for the problem when the lower and upper bounds coincide (Höppner & Klawonn (2008); Dinler & Tural (2016); de Maeyer et al. (2023); Lin et al. (2019); Ganganath et al. (2014); Chakraborty & Das (2019); Tang et al. (2019)). However, forcing the upper and lower bounds to be exactly equal is highly impractical since in most settings only lower and upper bounds are desired. Lei et al. (2013) provide heuristics for $k$-means clustering with **EL** constraints. Approximation algorithms have been obtained for clustering objectives other than the $k$-median and $k$-means. For example, Friggstad et al. (2016) gave an approximation algorithm for Facility location[6] with **EL** constraints violating both the bounds by a constant factor with a trade-off in them whereas Gupta et al. (2021) gave an approximation algorithm violating the upper bounds by a factor of $5/2$. For $k$-center with **EL** constraints, Ding et al. (2017) and Rösner & Schmidt (2018) independently gave constant factor approximations. To the best of our knowledge, the $k$-median problem with **EL** constraints has not been studied before in the literature.

Some prior works in fair clustering bear some resemblance to our work. For example, in settings where the points in the dataset belong to different demographic groups, the works of Chierichetti et al. (2017); Bercea et al. (2019); Bera et al. (2019); Esmaeili et al. (2020); Ahmadian et al. (2019) have considered a fairness notion where each cluster is constrained to have close to population-level proportions of each group. While this notion is similar to ours, there is a considerable difference since the bounds are not imposed on the cluster sizes as we do, but rather the proportions of the groups in each cluster. Further, another notion in fair clustering imposes lower and upper bounds not on the proportions of the demographic groups in each cluster but on the number of centers selected from each demographic group (Kleindessner et al. (2019); Jones et al. (2020); Hotegni et al. (2023)), i.e., in a dataset that consists of 50% from a "blue" group and 50% from a "red" group,[7] then if we cluster with $k = 10$, it may be desired to have at least 3 centers selected from each group and at most 7 centers from one group, thereby ensuring both a measure of diversity and restricted dominance in the selected centers. However, this notion is also different from the **EL** notion. Interestingly, Dickerson et al. (2024) present a modular approach to combine both demographic fairness notions mentioned above simultaneously. Although the above-mentioned demographic notions are different from imposing lower and upper bounds on the cluster sizes, the objective of Dickerson et al. (2024) of combining two notions simultaneously is similar to our objective. Further, their modular approach of post-processing existing solutions is also similar to our approach, although at a high level, our constraints and techniques are very different.

---

[6]In the facility location problem, instead of a hard bound $k$ on the number of facilities, every facility has a facility opening cost and the goal now is to minimize the total cost of opening a subset of facilities and serving the clients from these opened facilities.

[7]We denote the demographic groups with colors as done in fair clustering papers. Concretely, these colors could denote attributes such as age, gender, or race.

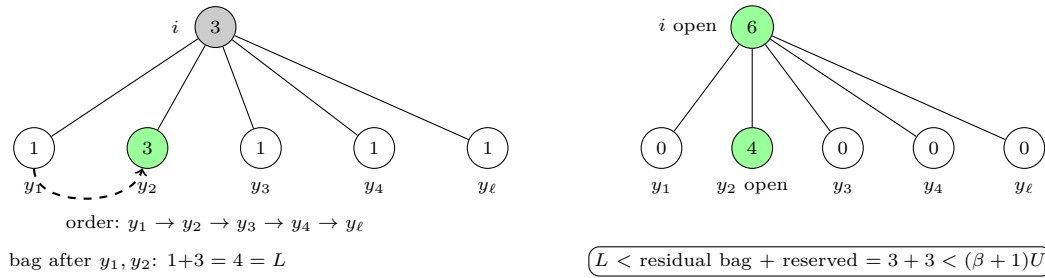

Figure 1: Toy examples for one star with $L = 4$, $U = 5$ and $\beta = 1$; the numbers in solid (grey) circle represents the reserved clients at star center $i$. Numbers in circles at the spokes represents unsettled unreserved clients. A bag of unsettled clients is accumulated while scanning spokes in decreasing distance from the center $i$. Whenever the bag first reaches $L$, the current spoke opens (except for last spoke). At the end, the center $i$ opens for all the remaining clients at last spoke along with the reserved clients.

## 5   High Level Idea of Our Algorithm

Our approach begins by decomposing the EL Clustering instance into two simpler subproblems: ($i$) A *lower-bounded k-median* instance $I_L$, obtained by dropping the upper bounds, and ($ii$) An *upper-bounded k-median* instance $I_U$, obtained by dropping the lower bounds. We solve ($i$) and ($ii$) using any available algorithms (approximate or heuristic), obtaining solutions $S_L$ and $S_U$ respectively.

Before presenting the details of the combination algorithm, we introduce the following key terms that will be used throughout the algorithmic descriptions.

**Star:**   A subgraph formed by a facility $i$ (the *star-center*) in $S_L$ and a set of facilities from the $S_U$ (its *spokes*) for which $i$ is the closest facility among all facilities in the lower-bounded solution.

**Open facility:**   A facility selected in the final EL Clustering solution to serve clients.

**Closed facility:**   A facility not selected in the final EL Clustering solution.

We now describe our combination algorithm.

**Grouping facilities into Stars:** We group facilities into *stars*. For each facility $i$ opened in the solution $S_L$, we form a star $S_i$ consisting of $i$ (the star-center) and all facilities $i'$ opened in the solution $S_U$ whose closest $S_L$ facility is $i$.

**Processing stars: Opening and Closing Facilities within a Star:** Within each star $S_i$, we consider its spokes in decreasing order of distance from $i$. We maintain a "bag" collecting unsettled clients from $S_U$ assignments; when the bag size reaches the lower bound $L$, we open the current spoke facility (except for the last spoke) and assign all clients in the bag to it. For all the clients accumulated at the last spoke, we open $i$. To achieve claimed lower and upper bounds exactly enough ($\max\{0, L - |clients\ at\ last\ spoke|\}$), clients are reserved before processing the star and assigned to the facility $i$ when it is opened. Refer Figure 1 for a simple toy example.

**Determine Processing Order:** The naive approach of processing stars in arbitrary order can lead to client conflicts, where clients intended for one star-center in $S_L$ have already been assigned to spokes in another star. To avoid this, we build a *dependency graph* whose nodes are stars and whose edges represent shared clients between center of one star and the spokes of another star. We transform the built graph into an *almost-DAG* (removing cycles except self-loops) via a careful reassignment step. Processing stars in topological order ensures no premature client assignments.

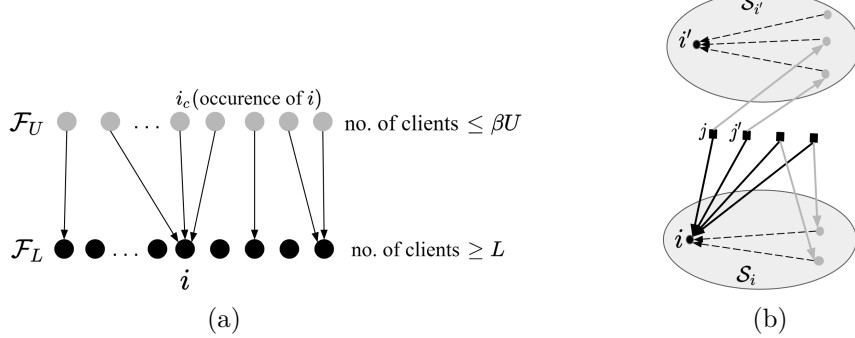

(a)                                          (b)

Figure 2: (a) Graph $G_1$: $I_L$ is an instance of **L$k$M** and $I_U$ that of **U$k$M**. (b) Let $L = 4$. Black and grey edges show the assignment of clients in $S_L$ and $S_U$, respectively. Star $\mathcal{S}_{i'}$ is processed before star $\mathcal{S}_i$. Clients $j, j'$ assigned to $i$ in $S_L$ have already been assigned to facilities in $\eta^{-1}(i')$ and hence are not available while processing $\mathcal{S}_i$.

## 6 Our Algorithm for EL Clustering

Let $I = (P, c, \mathcal{C}, \mathcal{F}, k, U, L)$ be an instance of **EL** Clustering. We first create an instance $I_L$ of **L$k$M** from $I$ by dropping the upper bounds and then an instance $I_U$ of **U$k$M** by dropping the lower bounds from $I$. Let $S_L = (\mathcal{F}_L, \sigma_L)$ and $S_U = (\mathcal{F}_U, \sigma_U)$ be solutions to $I_L$ and $I_U$, respectively. Let $\beta$ denote the violation in upper bounds, if any, in $S_U$. In the next section, we combine solutions $S_L$ and $S_U$ to obtain a solution $S_I = (\mathcal{F}_I, \sigma_I)$ to $I$ with $(\beta + 1)$ factor violation in upper bounds.

### 6.1 Combining solutions $S_L$ and $S_U$ to obtain $S_I$

To obtain a solution $S_I = (\mathcal{F}_I, \sigma_I)$ to $I$, we will open some facilities in $\mathcal{F}_L \cup \mathcal{F}_U$. We construct a directed graph $G_1$ on the set of facilities in $\mathcal{F}_L \cup \mathcal{F}_U$. For a facility $i \in \mathcal{F}_U$, let $\eta(i)$ denote the facility in $\mathcal{F}_L$ nearest to $i$ (assuming that the distances are distinct). Add an edge $(i, \eta(i))$ in the graph. Note that a facility $i$ may be open in both $S_L$ and $S_U$, in that case $i \in \eta^{-1}(i)$. In order to avoid self loops, when $i = \eta(i)$, we denote the occurrence of $i$ in $\mathcal{F}_U$ by $i_c$ so that $\eta(i_c) = i$. Thus, we obtain a forest of trees where-in each tree is a *star*. Formally, we define a star $\mathcal{S}_i$ to be a collection of nodes in $\{i\} \cup \eta^{-1}(i)$ with $i \in \mathcal{F}_L$ as the *star-center* and $\eta^{-1}(i) \subseteq \mathcal{F}_U$. See Figure 2-(a).

We process the stars to decide the set of facilities to open in $\mathcal{F}_L \cup \mathcal{F}_U$. Consider a star $\mathcal{S}_i$ centered at facility $i$. Clearly, the total assignments on $i$ in $S_L$ satisfy the lower bound but may violate the upper bound arbitrarily. On the other hand, the total assignments on a facility $i' \in \eta^{-1}(i)$ in $S_U$ satisfy the upper bound (within $\beta$ factor) but may violate the lower bound arbitrarily. We close some facilities in $\eta^{-1}(i)$ by transferring their clients (in $S_U$) to other facilities in $\eta^{-1}(i)$ if possible (or to $i$, if required) and open those at which the lower bound is satisfied. We may also have to open $i$ in the process. We make sure that upper bound is violated within the claimed bounds and the total number of facilities opened in $\mathcal{S}_i$ is at most $|\eta^{-1}(i)|$. The cardinality constraint is, hence, satisfied.

Suppose we consider the facilities in $\eta^{-1}(i)$ in the order of decreasing distance from $i$. Let the order be $y_1, y_2, ..., y_l$. We wish to collect the clients assigned to them, by $S_U$, in a bag looking for a facility $t$ at which we would have collected at least $L$ clients so that we can open $t$, empty the bag by assigning all the clients in the bag to $t$ and start the process again with the next facility in the order. The problem occurs when at the last facility ($y_l$), in the order, the bag has less than $L$ clients. In this case, we would like to assign these clients to the star-center $i$ making use of the fact that $i$ was assigned at least $L$ clients in $S_L$. The problem here is that the clients assigned to $i$ in $S_L$ might have been assigned to the facilities in $\eta^{-1}(i')$ for some star $\mathcal{S}_{i'}$ processed earlier or to the facilities in $\eta^{-1}(i)$ itself. Figure 2-(b) explains the situation. Thus, we need to process the stars in a carefully chosen sequence so as to avoid this kind of dependency amongst them. That is, the stars should be processed in such a way that if, at any point of time, we are processing a star

$\mathcal{S}_i$, then the clients assigned to $i$ in $S_L$ are not assigned to facilities in $\eta^{-1}(i')$ in $S_U$ for a star $S_{i'}$ processed earlier. For this, we construct a weighted directed (dependency) graph $G_2$ (possibly with directed cycles) on stars and convert it into a directed acyclic graph (DAG) (except possibly for self-loops), before processing the stars. A topological ordering in the graph, then gives us the order in which the stars must be processed. We will denote the graph by $G_2(\sigma_L, \sigma_U)$ to show that it is a function of the assignments in $S_L$ and $S_U$.

The graph $G_2(\sigma_L, \sigma_U)$ has the stars $\{\mathcal{S}_i : |\eta^{-1}(i)| > 0\}$ as the vertices. Let $\mathcal{X}(i_1, i_2) = \{j \in \mathcal{C} : \sigma_U(j) = i' \in \eta^{-1}(i_2) \text{ and } \sigma_L(j) = i_1\}$ i.e., $\mathcal{X}(i_1, i_2)$ is the set of clients that are served by $i_1$ in $S_L$ and by some facility at the spoke of the star centered at $i_2$ in $S_U$. We include the directed edge $(\mathcal{S}_{i_1}, \mathcal{S}_{i_2})$ from star $\mathcal{S}_{i_1}$ to $\mathcal{S}_{i_2}$ if $|\mathcal{X}(i_1, i_2)| > 0$. Let $w(\mathcal{S}_{i_1}, \mathcal{S}_{i_2}) = |\mathcal{X}(i_1, i_2)|$ denote the weight on the edge $(\mathcal{S}_{i_1}, \mathcal{S}_{i_2})$. Refer to Algorithm 1 and Figure 3-$(a) - (c)$ for the construction of graph $G_2$. Initially, $\mathcal{X}(i_1, i_2) = \emptyset$ and $w(\mathcal{S}_{i_1}, \mathcal{S}_{i_2}) = 0$ for all pairs of stars $\mathcal{S}_{i_1}$ and $\mathcal{S}_{i_2}$ ($i_1$ may be same as $i_2$). If the resulting graph has no directed cycle except possibly the self-loops, we are done. The graph $G_2$ is an *almost-DAG*. A directed graph is called an *almost-DAG*, if the only cycles in it are self loops. However, if there are non-trivial directed cycles in the graph, we redefine the assignments in $S_L$ to obtain another solution $\hat{S}_L = (\mathcal{F}_L, \hat{\sigma}_L)$ to break the cycles. The dependency graph for $(\hat{\sigma}_L, \sigma_U)$ will then be an *almost-DAG*.

---

**Algorithm 1:** Constructing Graph $G_2(\sigma_L, \sigma_U)$

**Input** : Stars $\mathcal{S}_i : i \in \mathcal{F}_L$
**Output:** Weighted Directed Graph $G_2(\sigma_L, \sigma_U) = (V, E)$
1 $V \leftarrow \{\mathcal{S}_i : |\eta^{-1}(i)| > 0\}$, $E \leftarrow \emptyset$
2 **for** $j \in \mathcal{C}$ **do**
3     $i' \leftarrow \sigma_U(j)$, $i_1 \leftarrow \sigma_L(j)$, $i_2 \leftarrow \eta(i')$
4     $\mathcal{X}(i_1, i_2) \leftarrow \mathcal{X}(i_1, i_2) \cup \{j\}$
5     $w(\mathcal{S}_{i_1}, \mathcal{S}_{i_2}) \leftarrow w(\mathcal{S}_{i_1}, \mathcal{S}_{i_2}) + 1$
6 **for** *each pair of stars* $\mathcal{S}_{i_1}$, $\mathcal{S}_{i_2}$ **do**
7     **if** $w(\mathcal{S}_{i_1}, \mathcal{S}_{i_2}) > 0$ **then**
8        $E \leftarrow E \cup (\mathcal{S}_{i_1}, \mathcal{S}_{i_2})$

---

**Algorithm 2:** Breaking Cycles: Constructing an *almost-DAG* $G_2(\hat{\sigma}_L, \sigma_U)$

**Input** : Graph $G_2(\sigma_L, \sigma_U)$
**Output:** $G_2(\hat{\sigma}_L, \sigma_U)$
1 $\hat{\sigma}_L(j) \leftarrow \sigma_L(j) \ \forall j \in \mathcal{C}$
2 **while** $\exists$ *a directed cycle* $< \mathcal{S}_{i_1}, \mathcal{S}_{i_2}, \ldots, \mathcal{S}_{i_q} > (q > 1)$ *in* $G_2$ **do**
3     $\kappa \leftarrow w(\mathcal{S}_{i_1}, \mathcal{S}_{i_2})$ // assume $(\mathcal{S}_{i_1}, \mathcal{S}_{i_2})$ as the minimum weight edge in the cycle
4     **for** $r = 1$ *to* $q$ **do**
5        $count \leftarrow 0$, $s \leftarrow (r \mod q) + 1$
6        **for** $j \in \mathcal{X}(i_r, i_s)$ **do**
7           **if** $count < \kappa$ **then**
8              $\hat{\sigma}_L(j) \leftarrow i_s$
9              $count + +$
10              $\mathcal{X}(i_r, i_s) \leftarrow \mathcal{X}(i_r, i_s) \setminus \{j\}$
11              $\mathcal{X}(i_s, i_s) \leftarrow \mathcal{X}(i_s, i_s) \cup \{j\}$
12        $w(\mathcal{S}_{i_s}, \mathcal{S}_{i_s}) \leftarrow w(\mathcal{S}_{i_s}, \mathcal{S}_{i_s}) + \kappa$
13        $w(\mathcal{S}_{i_r}, \mathcal{S}_{i_s}) \leftarrow w(\mathcal{S}_{i_r}, \mathcal{S}_{i_s}) - \kappa$
14        **if** $w(\mathcal{S}_{i_r}, \mathcal{S}_{i_s}) = 0$ **then**
15           $E \leftarrow E \setminus (\mathcal{S}_{i_r}, \mathcal{S}_{i_s})$ // Remove edge $(\mathcal{S}_{i_r}, \mathcal{S}_{i_s})$ from $G_2$
16          $E \leftarrow E \cup (\mathcal{S}_{i_s}, \mathcal{S}_{i_s})$ // Add edge $(\mathcal{S}_{i_s}, \mathcal{S}_{i_s})$ in $G_2$

---

**Breaking the cycles**: For graph $G_2(\sigma_L, \sigma_U)$, let $SC = < \mathcal{S}_{i_1}, \mathcal{S}_{i_2}, \ldots, \mathcal{S}_{i_q} >$ be a non-trivial directed cycle with $q > 1$. Without loss of generality, let $(\mathcal{S}_{i_1}, \mathcal{S}_{i_2})$ be the minimum weight edge in the cycle. We reassign any $\kappa = w(\mathcal{S}_{i_1}, \mathcal{S}_{i_2})$ clients in $\mathcal{X}(i_r, i_{(r \mod q)+1})$ from $i_r$ to $i_{(r \mod q)+1}$, increment the weight of the edge

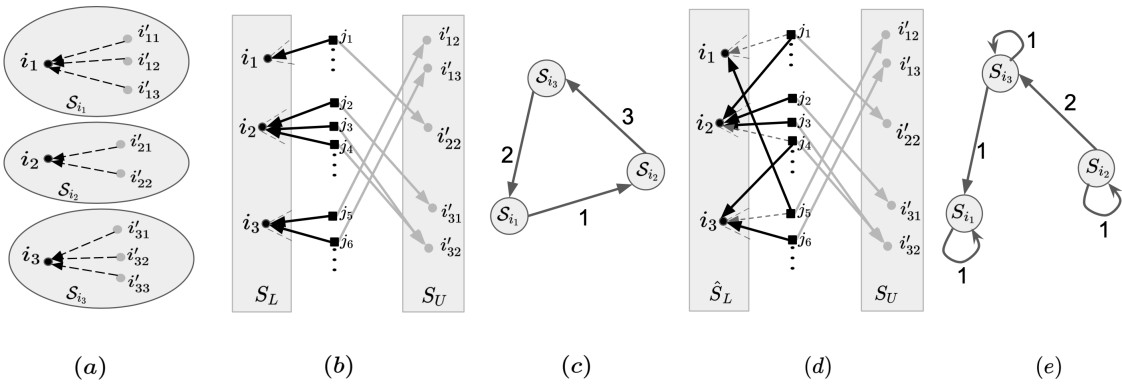

Figure 3: (a) Stars $\mathcal{S}_{i_1}$, $\mathcal{S}_{i_2}$ and $\mathcal{S}_{i_3}$; (b) $\mathcal{X}(i_1, i_2) = \{j_1\}$, $\mathcal{X}(i_2, i_3) = \{j_2, j_3, j_4\}$, $\mathcal{X}(i_3, i_1) = \{j_5, j_6\}$; (c) Its directed cycle $G_2(\sigma_L, \sigma_U)$; (d) Breaking a cycle: assign $j_1$ to $i_2$, $j_4$ to $i_3$ and $j_5$ to $i_1$, that is, $\hat{\sigma}_L(j_1) = i_2$, $\hat{\sigma}_L(j_4) = i_3$ and, $\hat{\sigma}_L(j_5) = i_1$; (e) The sub-graph $G_2(\hat{\sigma}_L, \sigma_U)$ after breaking the cycle.

$w(\mathcal{S}_{i_{(r \mod q)+1}}, \mathcal{S}_{i_{(r \mod q)+1}})$ by $\kappa$ and, reduce the weight of the edge $w(\mathcal{S}_{i_r}, \mathcal{S}_{i_{(r \mod q)+1}})$ by $\kappa$ for $r = 1 \ldots q$. Note that this adds new self-loops in the graph; however, no new non-trivial edge is added. Also, observe that $|\hat{\sigma}_L^{-1}(i)| = |\sigma_L^{-1}(i)|$ and hence $|\hat{\sigma}_L^{-1}(i)| \geq L$ is maintained for all $i \in \mathcal{F}_L$ after the reassignments. The weight of the edge $(\mathcal{S}_{i_1}, \mathcal{S}_{i_2})$ becomes zero and we remove it, thereby breaking the cycle. See Algorithm 2 and Figure 3-$(d) - (e)$. Note that a client $j$ gets reassigned at most once in all the cycles as during re-assignment, it moves its contribution from a non-trivial edge to a self-loop and not to any other non-trivial edge. Next, we bound the cost of solution $\hat{S}_L$ in the Lemma 6.1.

**Lemma 6.1.** *The cost, $Cost(\hat{S}_L)$, of solution $\hat{S}_L$ is bounded by $Cost(S_L) + 2Cost(S_U)$.*

*Proof.* Let $j \in \mathcal{C}$. The cost paid by $j$ in solution $\hat{S}_L$ is (see Figure 4-(a).): $c(j, \hat{\sigma}_L(j)) \leq c(j, \sigma_U(j)) + c(\sigma_U(j), \hat{\sigma}_L(j)) \leq c(j, \sigma_U(j)) + c(\sigma_U(j), \sigma_L(j)) \leq c(j, \sigma_U(j)) + (c(\sigma_U(j), j) + c(j, \sigma_L(j))) = c(j, \sigma_L(j)) + 2c(j, \sigma_U(j))$, where the second inequality holds since $\eta(\sigma_U(j)) = \hat{\sigma}_L(j)$. Summing over all $j \in \mathcal{C}$, we get the desired claim. $\square$

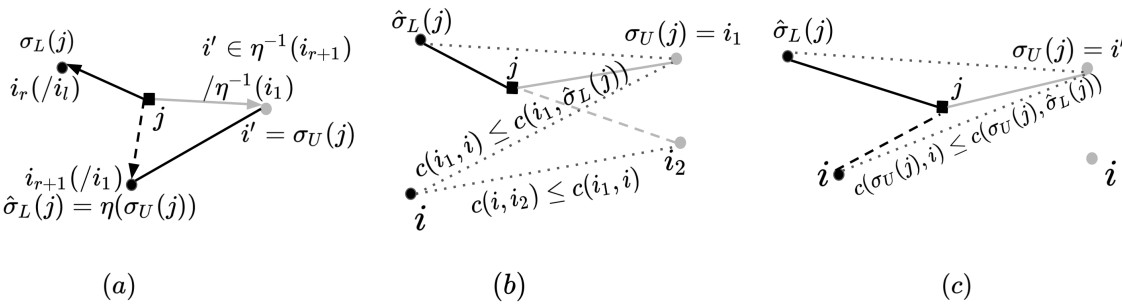

Figure 4: (a) $c(j, \hat{\sigma}_L(j)) \leq c(j, \sigma_L(j)) + 2c(j, \sigma_U(j))$; (b) Cost bound of *Type-I* assignments; (c) Cost bound of *Type-II* assignments.

Graph $G_2(\hat{\sigma}_L, \sigma_U)$ has the following properties:

1. $G_2(\hat{\sigma}_L, \sigma_U)$ is an *almost-DAG*.

2. $|\hat{\sigma}_L^{-1}(i)| \geq L \ \forall \ i \in \mathcal{F}_L$.

Now that we have an *almost-DAG* on the stars, we process the stars in the sequence $< \mathcal{S}_{i_1}, \mathcal{S}_{i_2}, \ldots \mathcal{S}_{i_t} >$ defined by a topological ordering of the vertices in $G_2(\hat{\sigma}_L, \sigma_U)$ (ignoring the self-loops). While processing the stars, we maintain partition of our clients into two sets, $\mathcal{C}_s$ and $\mathcal{C}_u$ of *settled* and *unsettled* clients respectively. We say that a client is *settled* if it has been assigned to an open facility in $S_I$ and *unsettled* otherwise. Initially $\mathcal{C}_s = \emptyset$ and $\mathcal{C}_u = \mathcal{C}$. As we process the stars, more and more clients get settled.

---

**Algorithm 3:** Process($\mathcal{S}_i$)

**Input** : $\mathcal{S}_i, i \in \mathcal{F}_L$

1   $reserved(i) \leftarrow \emptyset, Bag \leftarrow \emptyset$
2   **for** $i' \in \eta^{-1}(i)$ **do**
3     $\big|$   $N_{i'} \leftarrow \mathcal{C}_u \cap \sigma_U^{-1}(i')$
4   Arrange the facilities in $\eta^{-1}(i)$ in the sequence $< y_1, \ldots y_l >$ such that
    $c(y_{l'}, i) \geq c(y_{l'+1}, i) \ \forall \ l' = 1 \ldots l-1$
5   **if** $|N_{y_l}| < L$ **then**
6     $reserved(i) \leftarrow$ set of any $L - |N_{y_l}|$ clients from $\hat{\sigma}_L^{-1}(i) \setminus N_{y_l}$
7     **for** $i' \in \eta^{-1}(i)$ **do**
8       $\big|$   $N_{i'} \leftarrow N_{i'} \setminus reserved(i)$
9   **for** $l' = 1$ *to* $l-1$ **do**
10    $Bag \leftarrow Bag \cup N_{y_{l'}}$
11    **if** $|Bag| \geq L$ **then**
12      Open facility $y_{l'}$
13      **for** $j \in Bag$ **do**
14        $\big|$   Assign $j$ to $y_{l'}, \mathcal{C}_s \leftarrow \mathcal{C}_s \cup \{j\}, \mathcal{C}_u \leftarrow \mathcal{C}_u \setminus \{j\}$
15      $Bag \leftarrow \emptyset$
16   $t \leftarrow i$
17   **if** $|Bag \cup N_{y_l} \cup reserved(i)| > (\beta + 1)U$ **then**
18    $\big|$   $t \leftarrow y_l$
19   Open $t$
20   **for** $j \in Bag \cup N_{y_l} \cup reserved(i)$ **do**
21    $\big|$   Assign $j$ to $t, \mathcal{C}_s \leftarrow \mathcal{C}_s \cup \{j\}, \mathcal{C}_u \leftarrow \mathcal{C}_u \setminus \{j\}$

---

Consider star $\mathcal{S}_i$. Algorithm 3 gives the processing of $\mathcal{S}_i$ in detail. For $i' \in \eta^{-1}(i)$, let $N_{i'}$ be the set of unsettled clients, assigned to $i'$ in $S_U$. Consider the facilities in $\eta^{-1}(i)$ in decreasing order of distance from $i$, i.e., $y_1, y_2, ..., y_l$. We make sure that at most one of $y_l$ and $i$ is opened. To meet the lower bound at $i$ when $i$ is opened (and $y_l$ is closed if $y_l \neq i$), we reserve $\max\{0, L - |N_{y_l}|\}$ clients from $\hat{\sigma}_L^{-1}(i) \setminus N_{y_l}$ at $i$ (line 6). Observe that the topological ordering of the stars ensures that $|\hat{\sigma}_L^{-1}(i)| = |\mathcal{C}_u \cap \hat{\sigma}_L^{-1}(i)| \geq L$ and hence $|\hat{\sigma}_L^{-1}(i) \setminus N_{y_l}| \geq L - |N_{y_l}|$. We delete the reserved clients from $N_{i'}, i' \in \eta^{-1}(i)$ (lines $7-8$) before processing the facilities in $\eta^{-1}(i)$. In lines $9-15$, as we process the facilities in $\eta^{-1}(i)$, we collect the unsettled clients assigned to the facilities in $\eta^{-1}(i)$ by $S_U$ in a bag looking for a facility $t$ at which we have collected at least $L$ clients. We open $t$ and empty the bag by assigning all the clients in the bag to $t$ (called *Type-I assignment*) and start the process again with the next facility in the order.

To make sure that we do not open more than $|\eta^{-1}(i)|$ facilities in $\mathcal{S}_i$, we open only one of $i$ and $y_l$ for the remaining $(Bag \cup N_{y_l} \cup reserved(i))$ clients. This also ensures that we do not open $i$ more than once. We prefer to open $i$ and give all the remaining clients to $i$ because (as we will show later) the cost of assigning clients from $Bag \cup N_{y_l}$ to $i$ is bounded whereas we do not know how to bound the cost of assigning clients in $reserved(i)$ to $y_l$. However, in case, it leads to more than acceptable violation in the capacity at $i$, we open $y_l$ and assign the remaining $(Bag \cup N_{y_l} \cup reserved(i))$ clients to it. We show that $reserved(i)$ is empty in the latter case. Algorithm 4 summarizes our combination algorithm for constructing $S_I$ from $S_U$ and $S_L$.

## 6.2 Analysis

Recall that the assignments done in lines $9 - 15$ are *Type-I* assignments. Let the assignment of clients to facility $i$ when $t = i$ in lines $20 - 21$ be called as *Type-II assignments* and those to facility $y_l$ when $t = y_l$ be

---

**Algorithm 4:** Constructing $S_I$

---

**Input** : $< S_L = \mathcal{F}_L, \sigma_L >, < S_U = \mathcal{F}_U, \sigma_U >$
**Output:** $S_I$

**1** Construct graph $G_1 = < \mathcal{F}_L \cup \mathcal{F}_U, E >$ where $E = \{(i', \eta(i')) : i' \in \mathcal{F}_U\}$.
**2** Construct graph $G_2(\sigma_L, \sigma_U)$.
**3** Construct an *almost-DAG* $G_2(\hat{\sigma}_L, \sigma_U)$ from $G_2(\sigma_L, \sigma_U)$ using Algorithm 2.
**4** Obtain a topological ordering $< \mathcal{S}_{i_1}, \mathcal{S}_{i_2} \ldots \mathcal{S}_{i_t} >$ of stars in the *almost-DAG* $G_2(\hat{\sigma}_L, \sigma_U)$.
**5** **for** $r = 1$ *to* $t$ **do**
**6** $\quad$ Process $\mathcal{S}_{i_r}$ using Algorithm 3

---

called as *Type-III assignments*. To prove our main theorem, we need to show that in the obtained solution $S_I$, the lower bounds are respected, the upper bounds are violated by a factor of at most $(\beta + 1)$, the cost of the solution is bounded and the running time is $O(k^3 + n)$. We first prove that the lower bounds are respected at the opened facilities in Lemma 6.2.

**Lemma 6.2.** *Number of clients assigned to an open facility $i$ in $\mathcal{F}$ is at least $L$.*

*Proof.* We will bound the lower bounds for all three type of assignments separately.

1. Observe that the facilities opened by Algorithm 3 in line 12 (*Type-I* assignment) satisfy the lower bounds by design of the algorithm.

2. In *Type-II* assignments, the star-center $i$ satisfies the lower bound (if opened at line 19) as $|Bag \cup N_{y_l} \cup reserved(i)| \geq L$ where the inequality follows because $|reserved(i)| = \max\{0, L - |N_{y_l}|\}$.

3. In *Type-III* assignments, facility $y_l$ (if opened at line 19) also satisfies the lower bound as $|Bag \cup N_{y_l} \cup reserved(i)| > (\beta + 1)U \geq 2L$ because $U \geq L$ and $\beta \geq 1$.

$\square$

We next, show that the upper bounds are violated by a factor of at most $(\beta + 1)$ at the opened facilities in Lemma 6.3.

**Lemma 6.3.** *Number of clients assigned to an open facility $i$ in $\mathcal{F}$ is no more than $(\beta + 1)U$.*

*Proof.* We will bound the violations in the upper bounds for three type of assignments separately.

1. Consider the facilities in $\eta^{-1}(i)$. These facilities receive clients only in *Type-I* assignments (lines $13-14$). Note that for $l' = 2, ..., l-1$, we have $|Bag| < L$ just before line 10 and hence $|Bag| < L + \beta U$ (just after line 10) $\leq (1 + \beta)U$ because $L \leq U$. For $l' = 1$, $|Bag| = 0$ just before line 10 and hence $|Bag| \leq \beta U$ (just after line 10).

2. For *Type-II* assignments, the bound holds trivially because the star-center $i$ receives clients only when $|Bag| + |N_{y_l}| + |reserved(i)| \leq (\beta + 1)U$.

3. The maximum number of clients received by facility $y_l$ in *Type-III* assignments is, $|Bag| + |N_{y_l}| + |reserved(i)| = |Bag| + |N_{y_l}| + \max\{0, L - |N_{y_l}|\} = |Bag| + \max\{L, |N_{y_l}|\} \leq L + \beta U \leq (\beta + 1)U$.

$\square$

The next lemma (Lemma 6.4) bounds the cost of our solution $(S_I)$ in terms of cost of solution $S_U$ and $S_L$.

**Lemma 6.4.** *The cost of solution $S_I$ is bounded by $7Cost(S_U) + 2Cost(S_L)$.*

*Proof.* Consider a star $\mathcal{S}_i$.

1. *Type-I* assignments: Let $j \in \mathcal{C}$ be assigned to a facility $i_2 \in \eta^{-1}(i)$ in our solution and to $i_1 \in \eta^{-1}(i)$ in $S_U$ i.e., $i_1 = \sigma_U(j)$ and $i_2 = \sigma_I(j)$. The cost paid by $j$ is (see Figure 4-(b)):

$$
\begin{aligned}
c(i_2, \ j) &\leq c(i_1, \ j) + c(i_1, \ i) + c(i, \ i_2) \\
&\leq c(i_1, \ j) + 2c(i_1, \ i) \\
&\leq c(i_1, \ j) + 2c(i_1, \ \hat{\sigma}_L(j)) \qquad \text{(as } \eta(i_1) = i) \\
&= 3c(i_1, \ j) + 2c(j, \ \hat{\sigma}_L(j)) \qquad \text{(by triangle inequality)} \\
&\leq 2c(j, \ \sigma_L(j)) + 7c(j, \ \sigma_U(j)) \qquad \text{(by Lemma 6.1).}
\end{aligned}
$$

2. *Type-II* assignments: Let $j \in reserved(i)$ be assigned to $i$. Also, let $j \in N_{i'} : \ i' \in \eta^{-1}(i)$ be such that $i' = \sigma_U(j)$. Then, the cost (see Figure 4-(c)) is:

$$
\begin{aligned}
c(i, \ j) &= c(\sigma_U(j), \ j) + c(\sigma_U(j), \ i) \\
&= c(\sigma_U(j), \ j) + c(\sigma_U(j), \ \eta(\sigma_U(j))) \\
&= c(\sigma_U(j), \ j) + c(\sigma_U(j), \ \hat{\sigma}_L(j)) \\
&\leq c(\sigma_U(j), \ j) + \ c(\sigma_U(j), \ j) + c(j, \ \hat{\sigma}_L(j)) \\
&= 2c(j, \ \sigma_U(j)) + c(j, \ \hat{\sigma}_L(j)) \\
&\leq 4c(j, \ \sigma_U(j)) + c(j, \ \sigma_L(j))
\end{aligned}
$$

where the second and third equality follow because $\hat{\sigma}_L(j) = i = \eta(\sigma_U(j))$ and the last inequality follows by Lemma 6.1.

3. *Type-III* assignments: Note that $|Bag \cup N_{y_l} \cup reserved(i)| > (\beta + 1)U \Rightarrow |reserved(i)| = 0$, for otherwise $|N_{y_l} \cup reserved(i)| = L$ and thus $|Bag \cup N_{y_l} \cup reserved(i)| < L + L \leq 2U$ because $L \leq U$. Hence, the cost of assigning $|Bag \cup N_{y_l}|$ clients to $y_l$ is bounded in the same manner as the cost of *Type-I* assignments.

By summing the cost over all the assignments of *Type-I*, *Type-II* and *Type-III*, we get, $Cost(S_I) \leq 7Cost(S_U) + 2Cost(S_L)$ □

We finally show bounds on the running time of our algorithm in the following lemma.

**Lemma 6.5.** *Running time of our combination algorithm (Algorithm 4) is $O(k^3 + n)$.*

*Proof.* Constructing $G_1$ takes $O(k^2)$ time and the graph $G_2$ can be constructed in time $O(n + k^2)$: for each client $j$, one can determine the edge $(i_1, i_2)$ to which $j$ contributes in constant time. $G_2$ can be converted into almost-DAG in $O(k^3 + n)$ time using DFS and Algorithm 2: computing minimum weight edges takes at most $O(k^3)$ time over the entire algorithm and every client is re-assigned at most once. The time taken by Algorithm 3 when executed on all stars is no more than $O(n + k \log k)$; note that in this case also, a client is re-assigned at most once; $k \log k$ comes from sorting in step 4. Thus, having obtained solutions to L$k$M and U$k$M, combining the two solutions take $O(k^3 + n)$ time. □

Since any solution to $I$ is feasible for $I_L$ and $I_U$, we have $Cost(S_L) \leq Cost(S_I)$ and $Cost(S_U) \leq Cost(S_I)$. Therefore, the proof of Theorem 3.1 follows from Lemmas 6.2, 6.3, 6.4 and, 6.5. Furthermore, we have $Cost(O_L) \leq Cost(O)$ and $Cost(O_U) \leq Cost(O)$, where $O, O_L$ and $O_U$ denote optimal solution to $I, I_L$ and $I_U$, respectively. Therefore, the proof of Corollary 3.2 follows from Theorem 3.1 and by using approximation algorithms such as Byrka et al. (2016) for **U$k$M** to obtain $S_U$ and Han et al. (2020a) for **L$k$M** to obtain $S_L$.

## 7 Conclusion and Future Work

In this paper, we presented a modular approach for solving the **EL** Clustering problem by combining a solution of the $k$-median problem where the cluster sizes are lower bounded with another where the cluster sizes are upper bounded. Our solution introduces a bounded degradation over the costs of the given solutions.

Further, given a solution to the upper bounded instance where the upper bounds are violated by $\beta$ our solution only incurs a bounded additional violation leading to at most a $\beta + 1$ violation. An advantage of our method is that it gains from any improvements in the upper bounded and lower bounded solutions. Specifically, solutions for the upper and lower bounded instances with better approximation ratios enable us to obtain solutions with a better approximation ratio for **EL**. A similar note follows for solutions with smaller $\beta$ violations in the upper bound. Interestingly, we note that Lemma 6.2 and Lemma 6.3 and hence our results hold for a more general scenario where the lower and upper bounds are not necessarily the same across the facilities, the only restriction is that $\max_{i \in \mathcal{F}} L_i \leq \min_{i \in \mathcal{F}} U_i$. Furthermore, we discussed how our algorithm can be applied to other clustering variants including $k$-means clustering. Moreover, for the special case when the gap between the lower and upper bounds is large enough (specifically, $2L_i \leq U_i, \forall i \in \mathcal{F}$) the violation in the upper bound can be reduced to $\beta + \epsilon$ for a given $\epsilon > 0$.

One direction for future work would be to get rid of the plus 1 violation in the upper bounds. Another interesting direction is to extend the results general lower and upper bounds. We acknowledge that the constants associated with the cost of generating a $k$-means clustering with equitable load are rather high in our paper. Improving these constants is another useful direction for future work.

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

# A    Reducing violation in upper bounds when $2L \leq U$

---

**Algorithm 5:** $\text{Process}(\mathcal{S}_i)$

---

**Input**   : $\mathcal{S}_i : i \in \mathcal{F}_L$

**1** $reserved(i) \leftarrow \emptyset$, $Bag \leftarrow \emptyset$

**2** **for** $i' \in \eta^{-1}(i)$ **do**

**3** $\quad$ $N_{i'} \leftarrow \mathcal{C}_u \cap \sigma_U^{-1}(i')$

**4** Arrange the facilities in $\eta^{-1}(i)$ in the sequence $< y_1, \ldots y_l >$ such that
$\quad$ $c(y_{l'}, \ i) \geq c(y_{l'+1}, \ i) \ \forall \ l' = 1 \ldots l - 1$

**5** **if** $|N_{y_l}| < L$ **then**

**6** $\quad$ $reserved(i) \leftarrow$ set of any $L - |N_{y_l}|$ clients from $\hat{\sigma}_L^{-1}(i) \setminus N_{y_l}$

**7** $\quad$ **for** $i' \in \eta^{-1}(i)$ **do**

**8** $\quad\quad$ $N_{i'} \leftarrow N_{i'} \setminus reserved(i)$

**9** $Prev \leftarrow null$, $Prev_{count} = 0$

**10** **for** $l' = 1$ *to* $l - 1$ **do**

**11** $\quad$ $Bag \leftarrow Bag \cup N_{y_{l'}}$

**12** $\quad$ **if** $|Bag| \geq L$ **then**

**13** $\quad\quad$ Open facility $y_{l'}$

**14** $\quad\quad$ $Count \leftarrow 0$

**15** $\quad\quad$ **for** $j \in Bag$ **do**

**16** $\quad\quad\quad$ **if** $Count \leq \beta U$ **then**

**17** $\quad\quad\quad\quad$ Assign $j$ to $y_{l'}$, $\mathcal{C}_s \leftarrow \mathcal{C}_s \cup \{j\}$, $\mathcal{C}_u \leftarrow \mathcal{C}_u \setminus \{j\}$, $Bag \leftarrow Bag \setminus \{j\}$, $Count++$

**18** $\quad$ **else**

**19** $\quad\quad$ $Prev \leftarrow y_{l'}$ // *Prev* denotes the last unopened facility in $\eta^{-1}(i)$

**20** $\quad\quad$ $Prev_{count} = |Bag|$

**21** **if** $|Bag \cup N_{y_l} \cup reserved(i)| \leq (\beta + \epsilon)U$ **then**

**22** $\quad$ Open $i$

**23** $\quad$ **for** $j \in Bag \cup N_{y_l} \cup reserved(i)$ **do**

**24** $\quad\quad$ Assign $j$ to $i$, $\mathcal{C}_s \leftarrow \mathcal{C}_s \cup \{j\}$, $\mathcal{C}_u \leftarrow \mathcal{C}_u \setminus \{j\}$

**25** $\quad$ return

**26** **if** $|Bag \cup N_{y_l} \cup reserved(i)| \leq (\beta + \epsilon)U$ **then**

**27** $\quad$ Open $y_l$

**28** $\quad$ **for** $j \in Bag \cup N_{y_l} \cup reserved(i)$ **do**

**29** $\quad\quad$ Assign $j$ to $y_l$, $\mathcal{C}_s \leftarrow \mathcal{C}_s \cup \{j\}$, $\mathcal{C}_u \leftarrow \mathcal{C}_u \setminus \{j\}$

**30** $\quad$ return

**31** Open $Prev$ and $y_l$ // $|reserved(i)| = 0$ when $|Bag \cup N_{y_l} \cup reserved(i)| > (\beta + \epsilon)U$

**32** $Count \leftarrow 0$

**33** $Bag \leftarrow Bag \cup N_{y_l} \cup reserved(i)$

**34** **for** $j \in Bag$ **do**

**35** $\quad$ **if** $Count \leq L$ **then**

**36** $\quad\quad$ Assign $j$ to $Prev$, $\mathcal{C}_s \leftarrow \mathcal{C}_s \cup \{j\}$, $\mathcal{C}_u \leftarrow \mathcal{C}_u \setminus \{j\}$, $Bag \leftarrow Bag \setminus \{j\}$, $Count++$

**37** $\quad$ **else**

**38** $\quad\quad$ Assign all remaining clients in $Bag$ to $y_l$ and Break

---

In this section, assuming $2L \leq U$, we modify Algorithm 3 to obtain Algorithm 5 that reduces the violation in upper bounds from $(\beta + 1)$ to $(\beta + \epsilon)$ for a given $\epsilon > 0$. In particular, we present the following results:

**Theorem A.1.** *For $2L \leq U$, given a solution $S_U$ for UPPER BOUNDED $k$-MEDIAN (UkM) violating the upper bound by a factor of $\beta$ and a solution $S_L$ for LOWER BOUNDED $k$-MEDIAN (LkM). If the clustering costs of the solutions are $Cost(S_U)$ and $Cost(S_L)$, respectively. Then, a solution of cost at most $(O(\frac{1}{\epsilon})(7Cost(S_U) + 2Cost(S_L)))$ can be obtained for **EL** Clustering that violates the upper bound by a factor of $(\beta + \epsilon)$ for a fixed $\epsilon > 0$.*

We do the following modifications to Algorithm 3: ($i$) on arriving at a facility, say $t$, at which $|Bag| \geq L$, we open $t$ and instead of emptying the bag, we assign only $\beta U$ clients to $t$. Remaining clients are carried forward to the next facility in the order; ($ii$) we keep account of the last facility (in $Prev$), if any, that is not opened, and the number of clients in the bag at that instant (in $Prev_{count}$) i.e., $Prev$ is the facility $y_{l'}$ for which $|Bag| < L$ immediately after line 14 (hence at line 23) and $Prev_{count} = |Bag|$ at that time. We open $Prev$ at the end, if required. This is done as follows: if $|Bag \cup reserved(i) \cup N_{y_l}| \leq (\beta + \epsilon)U$, we are done (we open $i(/y_l)$ and assign all clients to it). Else, we open both $Prev$ and $y_l$ (at line 42) (note that $Prev \neq y_l$ must exist in this case) and, distribute the clients in $Bag \cup reserved(i) \cup N_{y_l}$ among $Prev$ and $y_l$, so that they receive at least $L$ clients. We will show that the service costs and the violation in upper bounds are bounded in this case.

Let the assignment of clients to facility $Prev$ in line 47 be called as *Type-IV assignments*. The assignments in line 20, line 31 and lines 38 & 49 are *Type-I*, *Type-II* and *Type-III* assignments respectively. Before we proceed to prove our claims, note that we open at most one of $y_l$ and $i$: if $i$ is opened at line 29, we return at line 33 and thus $y_l$ is never opened in this case. As before, this ensures that $i$ is not opened more than once.

Clearly, lower bound is satisfied by *Type-I* and *Type-IV* assignments done in line 20 and 47 for the facilities opened in lines 16 and 42 respectively. Also, since $|Bag \cup N_{y_l} \cup reserved(i)| \geq L$, lower bound is satisfied by *Type-II* assignments done in line 31 for the facility $i$ opened in line 29. For *Type-III* assignments done at line 38, $|Bag \cup N_{y_l} \cup reserved(i)| > (\beta + \epsilon)U \geq (\beta + \epsilon)L$. Clearly, the upper bound is violated by a factor of at most $(\beta + \epsilon)$ at the facilities opened in lines 16, 29, 36 and $Prev$ in line 42. For the assignments done in line 49, we look at the status at line 42: $|reserved(i)| = 0$, for otherwise $|N_{y_l} \cup reserved(i)| = L$, hence $|Bag \cup reserved(i) \cup N_{y_l}| < L + L \leq U$. Thus, $|Bag \cup N_{y_l} \cup reserved(i)| = |Bag \cup N_{y_l}| < L + \beta U$. Also, $|Bag \cup N_{y_l} \cup reserved(i)| > (\beta + \epsilon)U > 2L$. Thus, $L < |Bag \cup N_{y_l} \cup reserved(i)| - L < \beta U$ i.e., at line 49, $L < |Bag| < \beta U$.

Costs of *Type-I*, *Type-II* and *Type-III* assignments are bounded in the same manner as in Section 6. To bound the service cost of *Type-IV* assignments (line 47), observe that $|Bag \cup reserved(i) \cup N_{y_l}| > (\beta + \epsilon)U \Rightarrow |Bag| > \epsilon U$ as $|reserved(i)| = 0$ and $|N_{y_l}| \leq \beta U$; hence, $Prev_{count} \geq |Bag|$(at line 25) $> \epsilon U > \epsilon L$. Note that $Prev$ and $Prev_{count}$ do not change after exiting the for-loop at line 27. Thus, $Prev_{count} > \epsilon L$ after line 42 also and the cost of assigning at most $L$ clients from $N_{y_l}$ to $Prev$ is bounded by $(1/\epsilon)$ times the cost of assigning $\epsilon L$ clients from $\cup_{i' \text{ occurs before } Prev \text{ in the order}} N_{i'} \cup N_{Prev}$ to $y_l$. Hence, the total cost of *Type-IV* assignments is bounded by $(1/\epsilon)$ total cost of *Type-III* assignments.

# B Modifications for Other related Problems

## B.1 $k$-Means with Equitable Load

The $k$-means problem with **EL** constraints is same as the $k$-median problem with **EL** constraints except that the goal now is to minimize the sum of the squared distances instead of minimizing the sum of distances from the assigned facilities. Further, the facilities to be selected in the $k$-means problem possibly belong to an infinite space. Note that, in the $k$-means problem,

1. the distances are squared which may not satisfy triangle inequality but they satisfy $\alpha$-relaxed triangle inequality, that is, $c(x, y) \leq \alpha c(x, z) + \alpha c(z, y)$ for $\alpha = 2$ and,

2. we can assume the set of facilities to be in a finite space by losing $2\alpha$ factor in the distances for $\alpha = 2$.

We create an instance $I_L$ of LOWER BOUNDED $k$-MEANS and $I_U$ of UPPER BOUNDED $k$-MEANS instead of **L**$k$**M** and **U**$k$**M**. Solution $S_I$ is obtained by using the combination algorithm on $S_U$ and $S_L$. Note that, the violation in upper bounds remains the same, that is, for $\beta$ violation in upper bounds in $S_U$, we get $(\beta + 1)$ violation in upper bounds in $S_I$. We next bound the cost of the obtained solution $S_I$. With relaxed triangle inequality, Lemma 6.1 can be modified to bound the cost, $Cost(\hat{S}_L)$, of solution $\hat{S}_L$ by $4Cost(S_L) + 6Cost(S_U)$.

**Lemma B.1.** *The cost of solution $S_I$ is bounded by $352Cost(S_U) + 192Cost(S_L)$.*

*Proof.* We will modify the proof Lemma 6.4 to accommodate relaxed triangle equality. Due to space constraints, We will give details of *Type-I* assignments which have the dominating cost. Cost of *Type-II* can be bounded by $30c(j, \sigma_U(j)) + 16c(j, \sigma_L(j))$ in a similar manner. Cost of *Type-III* assignments is same as *Type-I* assignments.

*Type-I* assignments: Consider a star $\mathcal{S}_i$. Let $j \in \mathcal{C}$ be assigned to a facility $i_2 \in \eta^{-1}(i)$ and to $i_1 \in \eta^{-1}(i)$ in $S_U$ i.e., $i_1 = \sigma_U(j)$ and $i_2 = \sigma_I(j)$. The cost paid by $j$ is: $c(i_2, j) \le \alpha \cdot c(i_2, i) + \alpha \cdot c(i, j) \le \alpha \cdot c(i_2, i) + \alpha^2 \cdot (c(i, i_1) + c(i_i, j)) \le \alpha^2 \cdot c(i_1, j) + (\alpha + \alpha^2)c(i_1, i) \le \alpha^2 \cdot c(i_1, j) + (\alpha + \alpha^2)c(i_1, \hat{\sigma}_L(j)) \le \alpha^2 \cdot c(i_1, j) + (\alpha^2 + \alpha^3) \cdot (c(i_1, j) + c(j, \hat{\sigma}_L(j))) = (2\alpha^2 + \alpha^3) \cdot c(i_1, j) + (\alpha^2 + \alpha^3) \cdot c(j, \hat{\sigma}_L(j)) = 16c(i_1, j) + 12c(j, \hat{\sigma}_L(j)) \le 88c(j, \sigma_U(j)) + 48c(j, \sigma_L(j))$, where the first, second, fourth inequality follow by relaxed triangle inequality, third inequality follows as $\eta(i_1) = i$, the last equality follows by setting value of $\alpha$ to 2 and the last inequality follows by bound on $Cost(\hat{S}_L)$.

We incur an additional multiplicative factor of $2\alpha$ due to the assumption that the points lie in a finite space. Multiplying by 4 for $\alpha = 2$, we get, $Cost(S_I) \le 352Cost(S_U) + 192Cost(S_L)$.

$\square$

## B.2 $k$-Center with Equitable Load

The $k$-CENTER problem with **EL** constraints is the same as the $k$-median with **EL** constraints except that the goal now is to minimize the maximum distance of a client from the assigned facility instead of minimizing the total distance. We create instance $I_L$ and $I_U$ of LOWER BOUNDED $k$-CENTER and UPPER BOUNDED $k$-CENTER respectively instead of **L$k$M** and **U$k$M**. Same bounds are obtained on the cost by taking the maximum of the cost of all the types of assignments. Bounds on violation in upper bounds remains the same.

## B.3 $k$-Facility Location with Equitable Load

The $k$-FACILITY LOCATION with **EL** constraints is a generalization of the $k$-median with **EL** constraints where for every facility $i \in \mathcal{F}$, we also have a facility opening cost $f_i$. The objective now is to identify $\mathcal{F}' \subseteq \mathcal{F}$ of size at most $k$ and an assignment $\sigma$ of clients to $\mathcal{F}'$ so as to minimize the sum of the distances of the clients from their assigned facilities plus the facility opening costs of the selected facilities. We create instance $I_L$ of LOWER BOUNDED $k$-FACILITY LOCATION by dropping the upper bounds and cardinality constraint. An instance $I_U$ of UPPER BOUNDED $k$-FACILITY LOCATION is created by dropping the lower bounds. We then follow the same procedure as described for $k$-median in Section 6.1 to combine the solutions of the two instances. Cost of assignment is bounded in the same manner. There is no loss in factor due to facility opening costs as we only open facilities in $(\mathcal{F}_L \cup \mathcal{F}_U)$. The violation in the upper bounds remains the same.

## B.4 Knapsack-Median with Equitable Load

KNAPSACK MEDIAN with **EL** constraints is another generalization of $k$-median with **EL** constraints where every facility $i$ has weight $f_i$ and instead of $k$, and we have a budget $B$ on the total weight. Therefore, the objective is to identify $\mathcal{F}' \subseteq \mathcal{F}$ and an assignment $\sigma$ of clients to $\mathcal{F}'$ so as to minimize the sum of the distances of the clients from the assigned facility subject to the constraint $\sum_{i \in \mathcal{F}'} f_i \le B$.

We first create an instance $I_U$ of UPPER BOUNDED KNAPSACK MEDIAN by dropping the lower bounds and instance $I_L$ of LOWER BOUNDED KNAPSACK MEDIAN from $I$ by dropping the upper bounds, reducing the set of facilities to $\mathcal{F}_U$ and setting budget to the budget of $S_U$ (note that this can be different from given budget $B$ if there is violation in budget in $S_U$, otherwise it is $B$ only). It can be shown that $Cost(O_L) \le (2 + Cost(S_U))Cost(O)$: if a client $j$ is assigned in the optimal solution $O$ to $I$, to a facility $i$ not in $\mathcal{F}_U$, we assign it to a facility $i'$, nearest to $i$, in $\mathcal{F}_U$. The cost $c(j, i') \le c(j, i) + c(i, i') \le c(j, i) + c(i, i'') \le$

$c(j,\ i) + c(j,\ i) + c(j,\ i'') = 2c(j,\ i) + c(j,\ i'')$ where $i'' \in \mathcal{F}_U : \sigma_U(j) = i''$ and the second inequality holds because $i'$ is nearest to $i$ and not $i''$.

We next use the same procedure as in Section 6.1 to combine solutions $S_L$ and $S_U$ of instances $I_L$ and $I_U$ respectively. Note that since $\mathcal{F}_L \subseteq \mathcal{F}_U$, for $i \in \mathcal{F}_L$, $y_l = i$ in the star $\mathcal{S}_i$. This is important to make sure that the total facility opening cost in our solution is no more than that of $S_U$ in case we open $i$.

### B.5  A summary of approximation results using current best $S_L$ and $S_U$

| Problem | Results for underlying problems | | Our results | Previous results |
|---|---|---|---|---|
| | $(\alpha_L)$ | $(\beta, \alpha_U)$ | $(\theta_U, \theta_{approx})$ | |
| $k$-Median | LFL (82.6) Ahmadian & Swamy (2013) | U$k$M $((1+\epsilon), O(1/\epsilon^2))$ Byrka et al. (2016) | $((2+\epsilon), O(1/\epsilon^2))$ | Nil |
| $k$-Center | L$k$C (2) Aggarwal et al. (2006) | U$k$C $(1, 6)$ Khuller & Sussmann (2000) | $(2, 46)$ | Nil |
| Facility Location | LFL (82.6) Ahmadian & Swamy (2013) | UFL $(1, 3)$ Aggarwal et al. (2013) | $(2, 186.2)$ | $(O(1), O(1))^{\#}$ Friggstad et al. (2016) |
| $k$-Facility Location | LFL (82.6) Ahmadian & Swamy (2013) | U$k$FL $((2+\epsilon), O(1/\epsilon^2))$ Grover et al. (2018) | $((3+\epsilon), O(1/\epsilon^2))$ | Nil |
| Knapsack Median* | LKnM (1608) Han et al. (2020a) | UKnM $((2+\epsilon), O(1/\epsilon^2))$ Grover et al. (2018) | $((3+\epsilon), O(1/\epsilon^2))$ | Nil |

Table 1: Our results. $\alpha_L$ and $\alpha_U$ indicate the approximation factor for the lower and upper bounded variant respectively of the problem under consideration. Also, $\theta_U$ and $\theta_{approx}$ indicate the violation in the upper bounds and the approximation factor respectively for the corresponding lower and upper bounded problem. 'Nil' represents that there is no result known for the problem. (*): the result violates the budget by a factor of $(1 + \epsilon)$. (#): the result violates the lower bounds also. Note that, to the best of our knowledge there is no known polynomial time approximation algorithm for upper bounded k-means problem.

## C  Experiments

We empirically evaluate our combination algorithm for the $k$-Median objective on benchmark datasets from the UCI Machine Learning Repository, as preprocessed in Almanza et al. (2022): **Adult**, **Diabetes**, and **Bank**. The **Adult** dataset Dua & Graff (2017) contains census records, the **Bank** dataset Moro et al. (2014) contains data points from a marketing campaign, and the **Diabetes** dataset Strack et al. (2014) contains admission records of diabetic patients. The input points are embedded in Euclidean space, and distances are measured using the $\ell_2$ metric. The algorithm is implemented in Python and executed on a Apple laptop with an Apple M1 processor, 8 GB RAM, running macOS (Tahoe 26 beta). All source code, logs, and charts are publicly available[8].

**Input solutions.**  The initial lower-bounded ($S_L$) and upper-bounded ($S_U$) solutions were obtained using greedy heuristics that respect the lower and upper bounds respectively; thus, $\beta = 1$. Since our combination algorithm is modular, it can operate on any feasible pair ($S_L, S_U$) and hence our focus is to evaluate the performance of the *combination step itself*, independent of the specific heuristics used to generate $S_L$ and $S_U$.

**Experimental design.**  We vary the number of clusters $k$ for $k \in \{10, 25, 50, 75, 100\}$. The lower bound $L$ and upper bound $U$ are derived from the natural cluster size $n/k$, with buffers of $\pm\{10\%, 25\%, 50\%\}$. Facility pool $\mathcal{F}$ of varying sizes ($\{5\%, 10\%, 20\%, 25\%\}$ of the input size) are selected randomly from the input points.

---

[8]https://github.com/0-rudra-0/el-clustering

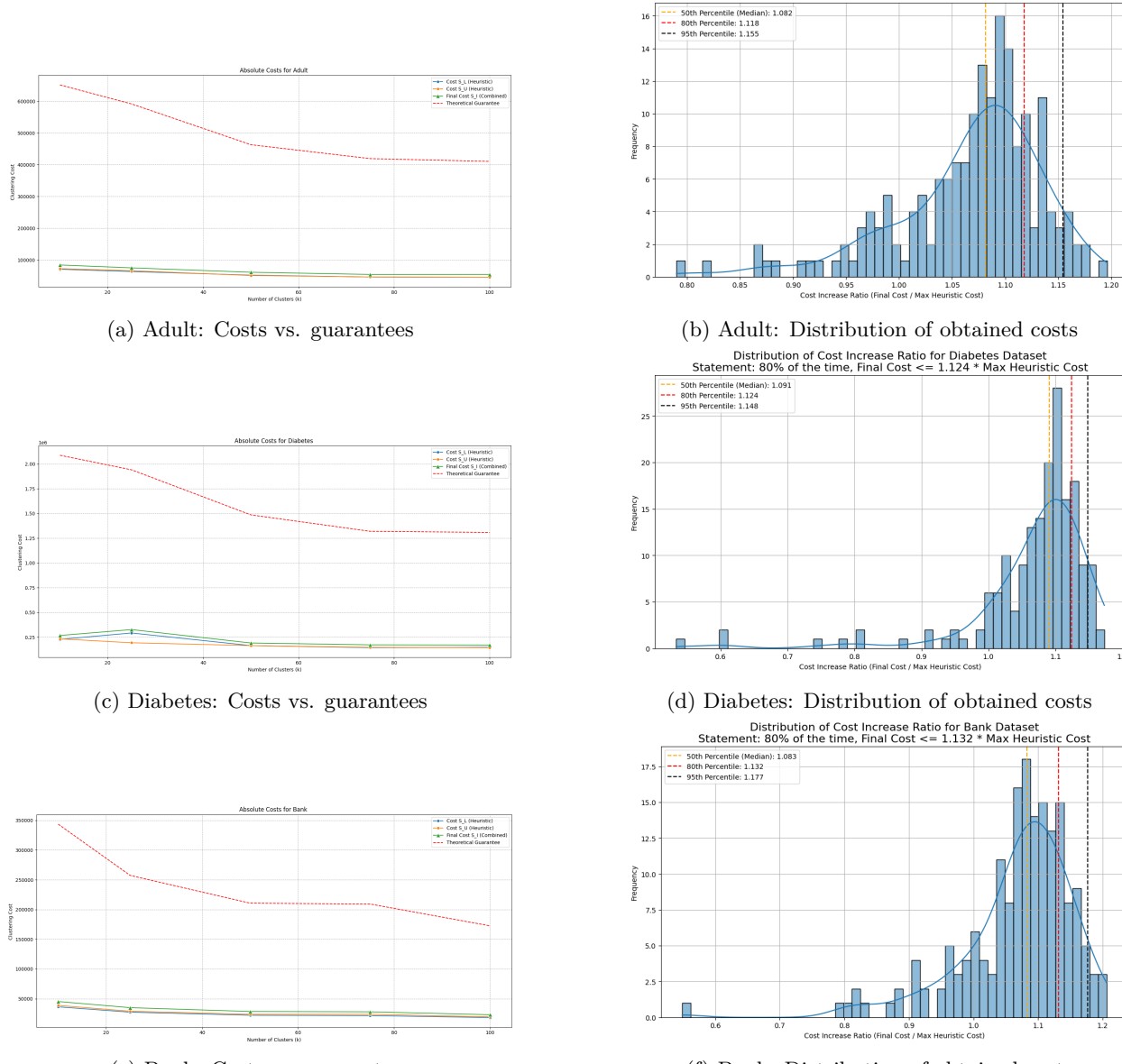

(a) Adult: Costs vs. guarantees

(b) Adult: Distribution of obtained costs

(c) Diabetes: Costs vs. guarantees

(d) Diabetes: Distribution of obtained costs

(e) Bank: Costs vs. guarantees

(f) Bank: Distribution of obtained costs

Figure 5: Left column (a, c, e) shows the plot of costs of $S_L$, $S_U$, our combined solution, and the theoretical guarantee ($7Cost(S_U) + 2Cost(S_L)$) against the values of $k$. For our combined solution, the $y$-axis reports the cost corresponding to the *worst observed gap* between the cost of our solution and $\max\{Cost(S_L), Cost(S_U)\}$. Right column (b, d, f): distribution of the ratio $\frac{Cost(\text{combined})}{\max\{Cost(S_L), Cost(S_U)\}}$ across all runs.

## C.1 Results and Insights

**Cost Analysis.** Figure 5 reports cost performance across all datasets. The left column (Figure 5 (a, c, e)) shows results as $k$ varies. For each value of $k$, we run the algorithm with different $(L, U)$ settings and varying the size of the facility pool. The $y$-axis reports the cost corresponding to the *worst observed gap* between the cost of our solution and $\max\{Cost(S_L), Cost(S_U)\}$. We observe that, across all datasets, our algorithm is close to the maximum of the input costs $Cost(S_L)$ and $Cost(S_U)$.

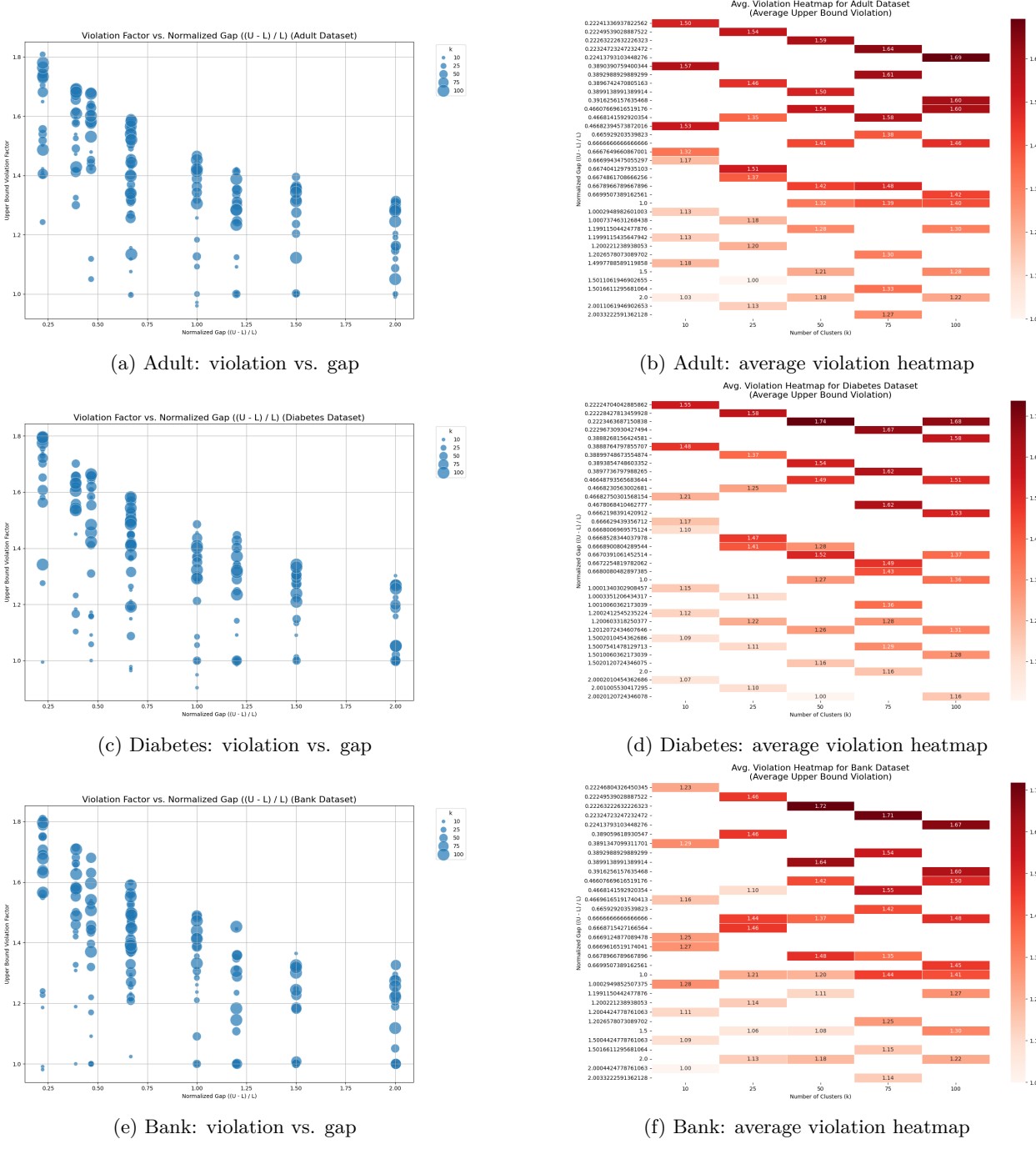

Figure 6: Left column (a, c, e): upper bound violation factor as a function of the $(U/L)$. Right column (b, d, f): heatmap showing the combined impact of $k$ and $(L, U)$ setting on the violation, on the cost averaged over varying sizes of the facility pool.

The right column (Figure 5 (b, d, f)) shows the full distribution of the ratio $\frac{Cost(\text{combined})}{\max\{Cost(S_L), Cost(S_U)\}}$ across all runs. The distribution is sharply concentrated near 1, confirming that the cost overhead is minimal in practice. In particular, the combined cost is within 1.2 of the cost of maximum of the two input solutions and it is typically 1.1.

Thus, the cost of our solution is substantially better than the worst-case theoretical bound.

**Upper Bound Violation.** Figure 6 analyzes the violation factor of the upper bound constraint. The left column (Figure 6 (a, c, e)) plots the violation factor as a function of the $U/L$ and $k$ for all sizes of the facility pool. It can be observed, the violation is within 1.54 in $\approx 80\%$ of the cases. The right column (Figure 6 (b, d, f)) presents a heatmap that shows the combined impact of $k$ and $(L, U)$ setting on the violation, on the cost averaged over varying sizes of the facility pool. It can be observed that the average violation over all the runs never exceeds 1.37 in all the data sets.

Further, we observed the trend that the performance degrades with higher values of $k$ and also with tight gap between $L$ and $U$. Since $k$ is typically small, violations in practice are expected to be small. The latter observation aligns with our theoretical analysis presented in Appendix A. Thus, in practice, violations remain well below the theoretical bound of $(\beta + 1)$ in most of the cases.

