# OpenReview forum: "EL-Clustering: Combining Upper- and Lower-Bounded Clusterings for Equitable Load Constraints"
_TMLR — Accepted by TMLR_

### Review · Reviewer_fkZg · 2025-07-26

**Summary Of Contributions:**

The paper develops an algorithm to solve the k-median clustering problem under both lower and upper bounds on cluster cardinalities. The proposed approach addresses the problem by combining solutions from two separate instances: one that omits the upper bound constraint (relaxes upper bound), and another that omits the lower bound constraint (relaxes lower bound).

**Audience:**

Yes

**Broader Impact Concerns:**

No.

**Claims And Evidence:**

No

**Requested Changes:**

As previously discussed, the paper requires substantial revisions before it can be considered for publication. If the following changes are implemented, the submission may become suitable for acceptance:
1.  Define key terms early: Clearly define terms such as star, star-center, spokes, open and closed facilities before they are used in the algorithmic descriptions.
2. Rewrite Section 3 (post-Corollary 3.2): Improve structure and clarity in this section. Present the approximation results and their implications more clearly, avoiding repetitive phrasing.
3. Substantially revise Section 5: This section is currently difficult to follow. Please rewrite with a clearer exposition, provide intuitive explanations of the algorithm, and ensure all figures are explicitly referenced and explained in the text.
4. Eliminate repetition: Reduce redundancy in the description of the modular strategy and citations throughout the paper.
4. Add empirical validation: Include experimental results (even synthetic or small-scale) to demonstrate the practical viability of the proposed approach.
5. Address the impact of upper bound violations and large approximation factors: Discuss when the approximation guarantees are acceptable in practice, and whether the violation factor $\beta+1$ can be mitigated in real-world scenarios.
6. Improve overall clarity and tone: Fix grammatical errors, avoid informal language, and ensure consistency in notation and definitions across the paper.

**Strengths And Weaknesses:**

Strengths:
1. The paper begins with a strong motivation, clearly outlining the practical relevance and importance of the equitable load (EL) clustering   problem.
2. The proposed solution adopts a modular strategy, which is noteworthy as it allows integration with a range of existing clustering algorithms.
3. The framework is generalizable beyond the k-median objective and can be extended to clustering formulations such as k-means, k-center, and knapsack median.

Weaknesses:
1. The final solution violates the upper bound constraint by a factor of β + 1, which may be undesirable in many real-world applications. Additionally, the overall approximation factor is quite large, limiting the method's practical usefulness.
2. The modular design makes the algorithm heavily dependent on the performance and quality of the subroutines used to solve the upper- and lower-bounded k-median problems.
3. The paper becomes significantly difficult to follow after Section 2. Section 3, particularly after Corollary 3.2, requires substantial rewriting due to lack of clarity and poor organization.
4. Section 5 is especially problematic: essential terms such as open facilities, closed facilities, stars, and spokes are used before being properly defined, making the remainder of the paper hard to follow.
5. There is noticeable repetition in citations and phrasing, especially regarding the modular approach, which detracts from readability and conciseness.
6. The paper lacks any empirical validation or experimental results to support its theoretical claims, making it difficult to assess its practical viability.
7. Overall, although the constrained clustering problem addressed in the paper is interesting and well motivated, the current presentation suffers from serious clarity and structural issues. The paper, in its present form, is not suitable for acceptance.

---

> ### Author Response · Authors · 2025-08-18
> **Rebuttal for Revision**
>
> Define key terms early:
>
> Response: We have added a dedicated “Definitions” paragraph in Section 5.
>
> Rewrite Section 3 (post-Corollary 3.2):
>
> Response: We have substantially reorganized the text following Corollary 3.2. The new version presents in different paragraphs: (i) a concise summary of the approximation guarantee, (ii) discussion on how the combination technique works irrespective of the technique used to solve underlying problems, (iii) a brief discussion of running time  and (iv) extension to other objectives and special case. Redundant phrases have been removed.
>
> Substantially revise Section 5:
>
> Response: Section 5 has been rewritten to focus on intuitive explanation of the algorithm. The revised section contains a high-level description of our algorithm’s overall strategy with clearly labeled paragraphs (e.g., grouping into Stars, processing stars and Processing Order via Dependency Graph).
> We have also added a simple toy example to explain the processing of a star to address the concern of another reviewer.
>
> Eliminate repetition:
>
> Response: The discussion of modular approach appears for the first time in introduction (before oragnization of the paper). We have removed the repeated explanation appearing in Section 3, paragraph 1– “We present a modular technique that combines the solutions of the lower bounded variant and the upper bounded variant of the problem to obtain our result stated in Theorem 3.1.” Again in the second-to-last paragraph of Section 3, the word “modular” is removed to decrease the redundancy.
> Repeated citations were a formatting issue for which changes similar to the ones shown below are done at various places. For example: Section 1 Para 2 third line: Chhabra et al. Chhabra et al. (2020)  and Cinà et al. Cinà et al. (2022)  —>  Chhabra et al. (2020)  and Cinà et al. (2022)
>
> Add empirical validation and address the impact of upper bound violations and large approximation factors:
>
> Response: Our paper is primarily theory-focused: the main contribution is in developing algorithms with provable guarantees. We would ideally like the work to stand on these theoretical results alone. That said, if the committee considers experiments essential, we are willing to include a small-scale evaluation on synthetic instances to illustrate the practical viability of our approach, though we would need some additional time to prepare this.
> Regarding the violation and approximation factors, we note that these are worst-case bounds and can be smaller in practice.
>
> Improve overall clarity and tone:
>
> Response: We have carefully edited the entire manuscript to improve grammar and tone. Informal expressions have been replaced with precise technical language. Notation is now consistent across all sections. Some of the changes are listed below:
>
> Abstract:
> In this paper —> In this paper,
>
> a slight violation to the upper bound —> a slight violation of the upper bound
>
> Our combination algorithms run   —> Our combination algorithm runs
>
> time where  n  …—> time, where n ….
>
> clustering objectives —> clustering objectives,
>
> Section 1:
>
> adversarial attacks —>  adversarial attacks,
>
> In fact, at least —> At least
>
> the community even  —> the community, even
>
> from the operations research point of view the selected centers in a clustering could represent facilities such as schools —> from the operations research point of view, the selected centers in a clustering could represent facilities such as schools,
>
> in revenue, at the same time the number of clients should not exceed a threshold —> in revenue; at the same time, the number of clients should not exceed a threshold,
>
> this notion which we call —> this notion, which we call
>
> same exact value —> same value
>
> Unlike the prior work —> Unlike the prior work,
>
> time where —> time, where
>
> $k$-median problem we —> $k$-median problem, we
>
> constraints including —> constraints, including
>
>
> Section 2:
>
> violates upper bound —> violates the upper bound
>
> clustering objective but —> clustering objective, but
>
> Section 3:
>
> these algorithms, the result —> these algorithms; the result
>
> the paper, modifications —> the paper; modifications
>
> Section 4:
>
> fair clustering bare  —> fair clustering bear
>
> population level proportions —> population-level proportions
>
> there is considerable difference —>  there is a considerable difference
>
> as we do but —> as we do, but
>
> $k=10$ it maybe —> $k=10$, it may be
>
> Although the above mentioned —> Although the above-mentioned
>
> our approach although —> our approach, although
>
> Section 5:
>
> introduce following —> introduce the following
>
> Subsection “Grouping facilities into Stars”:
>
> we form star—> we form a star
>
> Subsection “Processing stars: Opening and Closing Facilities within a Star:
>
> For all the client accumulated —> For all the clients accumulated
>
> Subsection “Determine Processing Order”:
>
>  spokes of another star -> the spokes of another star
>
> Revised pdf and summary of the changes will be uploaded in the revision of the paper.

---

### Review · Reviewer_eQhB · 2025-07-26

**Summary Of Contributions:**

The paper proposed a method of combining SU and SI clustering to find a EL clustering with a theoretical guarantee.

**Audience:**

Yes

**Broader Impact Concerns:**

No concern

**Claims And Evidence:**

Yes

**Requested Changes:**

- But, it is hard to read by nonexperts of constraint clustering.
  1. what is "violation factor"?
  2. Whar are $\alpha_U, \alpha_L$ approximation in Corollary 3.2?
  3. What ate 16-factor approximation, 3-factor violation, $O(1/\epsilon^2)$ approximation algorithm
- Can the problem be solved by linear programming?  This is typically done by fair clustering.
- It is hard to read Section 5 even though Section 5 explains the core idea of the proposed algorithm.
  1. What is 'Star" in Section 5?
  2. What is "yl - the last facility in the sequence"? What sequence is it?
  3. It would be good to explain the core idea with a toy example.

- Computation cost $O(k^3+n)$ is only for combination of SU and SL without
 including the cost for obtaining SU and SI. If it is true, this point should be mentioned.

- The main contribution of the proposed algorithm is to combine exsiting SU and SI algrithms to EL clustering.
  A better title would be " A novel algorithm to combine SU and SI clusterings to have EL clusterings" or something like that.

- In appendix B, for each method, it would be good to add references for SU and SI clustering algorithms.

**Strengths And Weaknesses:**

The paper is well written and the proposed method seems to be new and interesting and useful.
However, the paper is not easy to read for nonexperts in constraint clustering.

---

> ### Author Response · Authors · 2025-08-18
> **Rebuttal for Revision**
>
> Requested Changes:
>
> But, it is hard to read by nonexperts of constraint clustering.
> what is "violation factor"?
> Whar are  approximation in Corollary 3.2?
> What ate 16-factor approximation, 3-factor violation,  approximation algorithm
>
> Response: We note that the term “approximation factor” is standard terminology in the literature on approximation algorithms. Nevertheless, to avoid any ambiguity for non-experts, we have now explicitly defined both terms in the paper (Section 2, after the problem statement). In particular, we have added clear definitions for (i) approximation factor, describing the worst-case multiplicative gap to optimal cost, and (ii) violation factor, describing the permissible multiplicative relaxation of any of the constraints.
>
>
> Can the problem be solved by linear programming? This is typically done by fair clustering.
>
> Response: The standard LP relaxation for EL Clustering has an unbounded  integrality gap even if one of the bounds is allowed to be violated. However, strengthening techniques may be used to explore a possible solution.
>
>
> It is hard to read Section 5 even though Section 5 explains the core idea of the proposed algorithm.
> What is ‘’Star" in Section 5?
>
> Response: We have added a dedicated “Definitions” paragraph at the start of Section 5.
>
>
> What is "yl - the last facility in the sequence"? What sequence is it?
>
> Response: It was the same order referred to in the preceding paragraph. However to address the concern of another reviewer we have re-written this section and yl is no longer used.
>
>
> It would be good to explain the core idea with a toy example.
>
> Response: A simple  toy example is added in Section 5  to elaborate the processing of a star.
>
>
> Computation cost  is only for combination of SU and SL without including the cost for obtaining SU and SI. If it is true, this point should be mentioned.
>
> Response: This was mentioned in the paragraph after corollary 3.2  in context of comparison with results of underlying problems but now we have explicitly stated it after Theorem 3.1 as follows: “Note that the $O(k^3 + n)$ runtime stated above refers only to the combination step that merges the given solutions $S_U$ and $S_L$; it does not include the time required to compute $S_U$ and $S_L$, which depends on the algorithms used for UkM and LkM.”
>
>
> The main contribution of the proposed algorithm is to combine existing SU and SI algrithms to EL clustering. A better title would be " A novel algorithm to combine SU and SI clusterings to have EL clusterings" or something like that.
>
> Response: Thank you for the useful suggestion, we will change the title to {Combining Upper- and Lower-Bounded Clusterings for Equitable Load Constraints} in the camera ready.
>
>
> In appendix B, for each method, it would be good to add references for SU and SI clustering algorithms.
>
> Response: We have added a table summarizing the approximation guarantees obtainable for different problem variants by plugging in S_L and S_U from the respective problems, along with the relevant citations. See appendix B.
>
>
> Revised pdf and summary of the changes will be uploaded in the revision of the paper.

---

### Review · Reviewer_H7wK · 2025-08-02

**Summary Of Contributions:**

**The Problem Setting**:
An instance $I$ is defined by a set of facilities $F$, a set of clients $C$, metric cost $c(i,j)$ for serving a client $j \in C$ from a facility $i \in F$,  uniform lower bound $L$ on the minimum number of clients assigned to any open facility,  uniform upper bound $U$ on the maximum number of clients assigned to any open facility, and thethe maximum number of facilities that can be opened. The objective is to find the set of facilities $F'$ and assignments $\sigma: C \rightarrow F’$ that minimizes the total service cost.
$$
\min  \sum_{j \in C} c(j, \sigma(j))
$$
subject to
    $$
   |F'| \le k,  L \le |\sigma^{-1}(i)| \le U \quad \forall i \in F'.
    $$
 i.e., the number of opened facilities must be at most $k$, while also satisfying the lower and upper bound constraints for each opened facility $i \in F'$.

**Methods and Main Results**: From an instance $I$ of EL Clustering, we obtain the $\alpha_U$-approximation algorithms for Upper Bounded k-median (UkM)  and $\alpha_L$-approximation for Lower Bounded k-median (LkM). From these two solutions, the authors propose a non-trivial construction to obtain the approximation solutions for EL clustering, where violation of upper constraints guarantees $(\beta+1)$-factor and total-costs guarantees 7$\alpha_U$ + 2$\alpha_L$ factor. Time complexity is $O(k^3+n)$ time.

When integrating two solutions $S_U$ an $S_L$, which are derived from problems with separate constraints, a dependency graph of facility groups called stars is modeled (Algorithm 1). Cycles present in this graph are broken by redefining client assignments, thereby generating an
almost-DAG (Algorithm 2).
Finally, clients are reassigned according to a topological ordering of this almost-DAG in a non-strraigfoward procedures(Algorithms 3 and 4).

This process constructs an approximate solution that strictly preserves the lower constraints (Lemma 6.2) and only violates the upper constraints with factor $(\beta+1)$ (Lemma 6.3), while keeping the cost within a constant factor of the optimum (Lemma 6.1 and 6.4).

**Audience:**

Yes

**Broader Impact Concerns:**

N/ A

**Claims And Evidence:**

Yes

**Requested Changes:**

Questions

- Can the framework be extended to the non-uniform case, where
$ L_i$​ and $U_i$​ are different for each facility $i$?.


Request Changes:
- It would be great to add a table summarizes the approximation ratios and violation rates obtained in the paper for various problems.

**Strengths And Weaknesses:**

**Strengths**

- Clustering with both lower and upper bounds is important for practical applications. The lower bound constraint, in particular, is difficult to handle in facility location problems due to its combinatorial nature.
- The algorithm provides a bicriteria approximation guarantee for cost and constraint violation.
- The computational complexity of the algorithm is very low and practical.
- Constructing a solution that satisfies both lower and upper bounds from approximate solutions of problems with only one of these constraints is a highly non-trivial task. The algorithm's novelty lies in techniques such as constructing a directed graph and reassigning clients to break cycles.
-The framework can be extended to various similar types of clustering problems.

Weaknesses
- The bounds $L$ and $U$ are uniform. (Although this is still a difficult problem, so it may not be a major concern) .
- Regarding the writing, the results are presented as very general, which makes it difficult to see how much improvement was made over previous research for specific individual cases.

---

### Decision · Action_Editor_WVFW · 2025-09-19

**Recommendation:** Accept as is

**Additional Comments:**

As detailed above, the contribution is of interest and the claims are well substantiated.

**Audience:**

Yes

**Audience Explanation:**

This paper provides a useful way of combining two types of clustering algorithms - one obtaining a lower bound and one obtaining an upper bound - into one clustering algorithm that enjoys similar guarantees of both lower and upper bound. This is of theoretical interest and can also be useful in practice.

**Claims And Evidence:**

Yes

**Claims Explanation:**

After revision, the clarity of the definitions and the contributions has been improved, and the results seem well substantiated.